# Information Maximizing Curriculum:
## A Curriculum-Based Approach for Imitating Diverse Skills

**Denis Blessing** [*†] **Onur Celik**[†§] **Xiaogang Jia**[†‡] **Moritz Reuss**[‡]
**Maximilian Xiling Li**[‡] **Rudolf Lioutikov**[‡] **Gerhard Neumann**[†§]
[†] Autonomous Learning Robots, Karlsruhe Institute of Technology
[‡] Intuitive Robots Lab, Karlsruhe Institute of Technology
[§] FZI Research Center for Information Technology

## Abstract

Imitation learning uses data for training policies to solve complex tasks. However, when the training data is collected from human demonstrators, it often leads to multimodal distributions because of the variability in human actions. Most imitation learning methods rely on a maximum likelihood (ML) objective to learn a parameterized policy, but this can result in suboptimal or unsafe behavior due to the mode-averaging property of the ML objective. In this work, we propose *Information Maximizing Curriculum*, a curriculum-based approach that assigns a weight to each data point and encourages the model to specialize in the data it can represent, effectively mitigating the mode-averaging problem by allowing the model to ignore data from modes it cannot represent. To cover all modes and thus, enable diverse behavior, we extend our approach to a mixture of experts (MoE) policy, where each mixture component selects its own subset of the training data for learning. A novel, maximum entropy-based objective is proposed to achieve full coverage of the dataset, thereby enabling the policy to encompass all modes within the data distribution. We demonstrate the effectiveness of our approach on complex simulated control tasks using diverse human demonstrations, achieving superior performance compared to state-of-the-art methods.

## 1 Introduction

Equipping agents with well-performing policies has long been a prominent focus in machine learning research. Imitation learning (IL) [1] offers a promising technique to mimic human behavior by leveraging expert data, without the need for intricate controller design, additional environment interactions, or complex reward shaping to encode the target behavior. The latter are substantial advantages over reinforcement learning techniques [2, 3] that rely on reward feedback. However, a significant challenge in IL lies in handling the multimodal nature of data obtained from human demonstrators, which can stem from differences in preferences, expertise, or problem-solving strategies. Conventionally, maximum likelihood estimation (MLE) is employed to train a policy on expert data. It is well-known that MLE corresponds to the moment projection [4], causing the policy to average over modes in the data distribution that it cannot represent. Such mode averaging can lead to unexpected and potentially dangerous behavior . We address this critical issue by introducing *Information Maximizing Curriculum* (IMC), a novel curriculum-based approach.

In IMC, we view imitation learning as a conditional density estimation problem and present a mathematically sound weighted optimization scheme. Data samples are assigned curriculum weights, which are updated using an information projection. The information projection minimizes the reverse

---

*Correspondence to `denis.blessing@kit.edu`

KL divergence, forcing the policy to ignore modes it cannot represent [4]. As a result, the optimized policy ensures safe behavior while successfully completing the task.

Yet, certain modes within the data distribution may remain uncovered with a single expert policy. To address this limitation and endow the policy with the diverse behavior present in the expert data, we extend our approach to a mixture of expert (MoE) policy. Each mixture component within the MoE policy selects its own subset of training data, allowing for specialization in different modes. Our objective maximizes the entropy of the joint curriculum, ensuring that the policy covers all data samples.

We show that our method is able to outperform state-of-the-art policy learning algorithms and MoE policies trained using competing optimization algorithms on complex multimodal simulated control tasks where data is collected by human demonstrators. In our experiments, we assess the ability of the models to *i)* avoid mode averaging and *ii)* cover all modes present in the data distribution.

## 2   Preliminaries

Our approach heavily builds on minimizing Kullback-Leibler divergences as well as mixtures of expert policies. Hence, we will briefly review both concepts.

### 2.1   Moment and Information Projection

The Kullback-Leibler (KL) divergence [5] is a similarity measure for probability distributions and is defined as $D_{\text{KL}}(p\|p') = \sum_{\mathbf{x}} p(\mathbf{x}) \log p(\mathbf{x})/p'(\mathbf{x})$ for a discrete random variable $\mathbf{x}$. Due to its asymmetry, the KL divergence offers two different optimization problems for fitting a model distribution $p$ to a target distribution $p^*$ [4], that is,

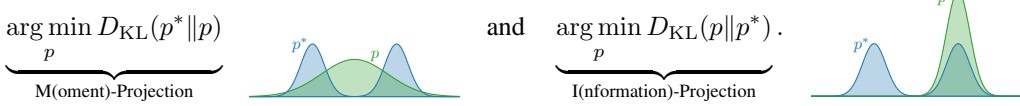

$$\underbrace{\arg\min_{p} D_{\text{KL}}(p^*\|p)}_{\text{M(oment)-Projection}} \quad \text{and} \quad \underbrace{\arg\min_{p} D_{\text{KL}}(p\|p^*)}_{\text{I(nformation)-Projection}}.$$

The M-projection - or equivalently maximum likelihood estimation (MLE) [6] - is *probability forcing*, meaning that the model is optimized to match the moments of the target distribution, causing it to average over modes that it cannot represent. In contrast, the I-projection is *zero forcing* which leads the model to ignore modes of the target distribution that it is not able to represent. The I-projection can be rewritten as maximum entropy problem, i.e.,

$$\arg\max_{p} \; \mathbb{E}_{p(\mathbf{x})}[\log p^*(\mathbf{x})] + \mathcal{H}(\mathbf{x}). \tag{1}$$

Using this formulation, it can be seen that the optimization balances between fitting the target distribution and keeping the entropy $\mathcal{H}(\mathbf{x}) = -\sum_{\mathbf{x}} p(\mathbf{x}) \log p(\mathbf{x})$ high.

### 2.2   Mixtures of Expert Policies

Mixtures of expert policies are conditional discrete latent variable models. Given some observation $\mathbf{o} \in \mathcal{O}$ and action $\mathbf{a} \in \mathcal{A}$, the marginal likelihood is decomposed into individual components, i.e.,

$$p(\mathbf{a}|\mathbf{o}) = \sum_{z} p(z|\mathbf{o})p(\mathbf{a}|\mathbf{o}, z).$$

Here, $\mathcal{O}$ and $\mathcal{A}$ denote the observation and action space respectively, and $z$ stands for the discrete latent variable that indexes distinct components within the mixture.

The gating $p(z|\mathbf{o})$ is responsible for soft-partitioning the observation space $\mathcal{O}$ into sub-regions where the corresponding experts $p(\mathbf{a}|\mathbf{o}, z)$ approximate the target density. Typically the experts and the gating are parameterized and learned by maximizing the marginal likelihood via expectation-maximization (EM) [7] or gradient ascent [8]. In order to sample actions, that is, $\mathbf{a}' \sim p(\mathbf{a}|\mathbf{o}')$ for some observation $\mathbf{o}'$, we first sample a component index from the gating, i.e., $z' \sim p(z|\mathbf{o}')$. The component index selects the respective expert to obtain $\mathbf{a}' \sim p(\mathbf{a}|\mathbf{o}', z')$.

# 3 Information Maximizing Curriculum

In this section, we propose Information Maximizing Curriculum (IMC), a novel algorithm for training mixtures of expert polices. We motivate our optimization objective using a single policy. Next, we generalize the objective to support learning a mixture of experts policy. Thereafter, we discuss the optimization scheme and provide algorithmic details.

## 3.1 Objective for a Single Expert Policy

We propose an objective that jointly learns a curriculum $p(\mathbf{o}, \mathbf{a})$ and a parameterized policy $p_{\boldsymbol{\theta}}(\mathbf{a}|\mathbf{o})$ with parameters $\boldsymbol{\theta}$. The curriculum is a categorical distribution over samples of a dataset $\{(\mathbf{o}_n, \mathbf{a}_n)\}_{n=1}^N$, assigning probability mass to samples according to the performance of the policy. To allow the curriculum to ignore samples that the policy cannot represent, we build on the I-projection (see Equation 1). We, therefore, formulate the objective function as

$$\tilde{J}(p(\mathbf{o}, \mathbf{a}), \boldsymbol{\theta}) = \mathbb{E}_{p(\mathbf{o}, \mathbf{a})}[\log p_{\boldsymbol{\theta}}(\mathbf{a}|\mathbf{o})] + \eta \mathcal{H}(\mathbf{o}, \mathbf{a}), \qquad (2)$$

which is optimized for $p(\mathbf{o}, \mathbf{a})$ and $\boldsymbol{\theta}$ in an alternating fashion using coordinate ascent [9]. We additionally introduce a trade-off factor $\eta$ that determines the pacing of the curriculum. For $\eta \to \infty$ the curriculum becomes uniform, exposing all samples to the policy and hence reducing to maximum likelihood estimation for $\boldsymbol{\theta}$. In contrast, if $\eta \to 0$ the curriculum concentrates on samples where the policy log-likelihood $\log p_{\boldsymbol{\theta}}(\mathbf{a}|\mathbf{o})$ is highest. The objective can be solved in closed form for $p(\mathbf{o}, \mathbf{a})$ (see Appendix B), resulting in

$$p^*(\mathbf{o}_n, \mathbf{a}_n) \propto p_{\boldsymbol{\theta}}(\mathbf{a}_n|\mathbf{o}_n)^{1/\eta}.$$

Maximizing the objective w.r.t $\boldsymbol{\theta}$ reduces to a weighted maximum-likelihood estimation, that is,

$$\boldsymbol{\theta}^* = \arg\max_{\boldsymbol{\theta}} \sum_n p(\mathbf{o}_n, \mathbf{a}_n) \log p_{\boldsymbol{\theta}}(\mathbf{a}_n|\mathbf{o}_n).$$

The optimization is repeated until reaching a local maximum, indicating that the curriculum has selected a fixed set of samples where the policy attains high log-likelihoods $\log p_{\boldsymbol{\theta}}(\mathbf{a}|\mathbf{o})$. Proposition 3.1 establishes convergence guarantees, the details of which are elaborated upon in Appendix A.1.

**Proposition 3.1.** *Let $\tilde{J}$ be defined as in Equation 2 and $0 < \eta < \infty$. Under mild assumptions on $p_{\boldsymbol{\theta}}$ and the optimization scheme for $\boldsymbol{\theta}$, it holds for all $i$, denoting the iteration index of the optimization process, that*

$$\tilde{J}(p(\mathbf{o}, \mathbf{a})^{(i+1)}, \boldsymbol{\theta}^{(i+1)}) \geq \tilde{J}(p(\mathbf{o}, \mathbf{a})^{(i)}, \boldsymbol{\theta}^{(i)}),$$

*where equality indicates that the algorithm has converged to a local optimum.*

The capacity to disregard samples where the policy cannot achieve satisfactory performance mitigates the mode-averaging problem. Nevertheless, a drawback of employing a single expert policy is the potential for significantly suboptimal performance on the ignored samples. This limitation is overcome by introducing multiple experts, each specializing in different subsets of the data.

## 3.2 Objective for a Mixture of Experts Policy

Assuming limited complexity, a single expert policy is likely to ignore a large amount of samples due to the zero-forcing property of the I-projection. Using multiple curricula and policies that specialize to different subsets of the data is hence a natural extension to the single policy model. To that end, we make two major modifications to Equation 2: Firstly, we use a mixture model with multiple components $z$ where each component has its own curriculum, i.e., $p(\mathbf{o}, \mathbf{a}) = \sum_z p(z)p(\mathbf{o}, \mathbf{a}|z)$. Secondly, we employ an expert policy per component $p_{\boldsymbol{\theta}_z}(\mathbf{a}|\mathbf{o}, z)$, that is paced by the corresponding curriculum $p(\mathbf{o}, \mathbf{a}|z)$. The resulting objective function is given by

$$J(\boldsymbol{\psi}) = \mathbb{E}_{p(z)}\mathbb{E}_{p(\mathbf{o}, \mathbf{a}|z)}[\log p_{\boldsymbol{\theta}_z}(\mathbf{a}|\mathbf{o}, z)] + \eta \mathcal{H}(\mathbf{o}, \mathbf{a}), \qquad (3)$$

where $\boldsymbol{\psi}$ summarizes the dependence on $p(z)$, $\{p(\mathbf{o}, \mathbf{a}|z)\}_z$ and $\{\boldsymbol{\theta}_z\}_z$. However, Equation 3 is difficult to optimize as the entropy of the mixture model prevents us from updating the curriculum of each component independently. Similar to [10], we introduce an auxiliary distribution $q(z|\mathbf{o}, \mathbf{a})$ to decompose the objective function into a lower bound $L(\boldsymbol{\psi}, q)$ and an expected $D_{\mathrm{KL}}$ term, that is,

$$J(\boldsymbol{\psi}) = L(\boldsymbol{\psi}, q) + \eta \mathbb{E}_{p(\mathbf{o}, \mathbf{a})} D_{\mathrm{KL}}(p(z|\mathbf{o}, \mathbf{a})\|q(z|\mathbf{o}, \mathbf{a})), \qquad (4)$$

with $p(z|\mathbf{o},\mathbf{a}) = p(\mathbf{o},\mathbf{a}|z)p(z)/p(\mathbf{o},\mathbf{a})$ and

$$L(\boldsymbol{\psi},q) = \mathbb{E}_{p(z)}\Big[\underbrace{\mathbb{E}_{p(\mathbf{o},\mathbf{a}|z)}[R_z(\mathbf{o},\mathbf{a})] + \eta\mathcal{H}(\mathbf{o},\mathbf{a}|z)]}_{J_z(p(\mathbf{o},\mathbf{a}|z),\boldsymbol{\theta}_z)}\Big] + \eta\mathcal{H}(z),$$

with $R_z(\mathbf{o}_n,\mathbf{a}_n) = \log p_{\boldsymbol{\theta}_z}(\mathbf{a}_n|\mathbf{o}_n,z) + \eta\log q(z|\mathbf{o}_n,\mathbf{a}_n)$, allowing for independent updates for $p(\mathbf{o},\mathbf{a}|z)$ and $\boldsymbol{\theta}_z$ by maximizing the per-component objective function $J_z(p(\mathbf{o},\mathbf{a}|z),\boldsymbol{\theta}_z)$. A derivation can be found in Appendix B.1. Since $\mathbb{E}_{p(\mathbf{o},\mathbf{a})}D_{\mathrm{KL}}(p(z|\mathbf{o},\mathbf{a})\|q(z|\mathbf{o})) \geq 0$, $L$ is a lower bound on $J$ for $\eta \geq 0$. Please note that the per-component objective function $J_z$ is very similar to Equation 2, with $J_z$ including an additional term, $\log q(z|\mathbf{o},\mathbf{a})$, which serves the purpose of preventing different curricula from assigning probability mass to the same set of samples: Specifically, a component $z$ is considered to 'cover' a datapoint $(\mathbf{o}_n,\mathbf{a}_n)$ when $q(z|\mathbf{o}_n,\mathbf{a}_n) \approx 1$. Since $\sum_z q(z|\mathbf{o}_n,\mathbf{a}_n) = 1$, it follows that for other components $z' \neq z$ it holds that $q(z'|\mathbf{o}_n,\mathbf{a}_n) \approx 0$. Consequently, $\log q(z'|\mathbf{o}_n,\mathbf{a}_n) \to -\infty$, implying that $R_{z'}(\mathbf{o}_n,\mathbf{a}_n) \to -\infty$. As a result, the other curricula effectively disregard the datapoint, as $p(\mathbf{o}_n,\mathbf{a}_n|z') \propto \exp R_{z'}(\mathbf{o}_n,\mathbf{a}_n) \approx 0$.

We follow the optimization scheme of the expectation-maximization algorithm [7], that is, we iteratively maximize (M-step) and tighten the lower bound (E-step) $L(\boldsymbol{\psi},q)$.

### 3.3 Maximizing the Lower Bound (M-Step)

We maximize the lower bound $L(\boldsymbol{\psi},q)$ with respect to the mixture weights $p(z)$, curricula $p(\mathbf{o},\mathbf{a}|z)$ and expert policy parameters $\boldsymbol{\theta}_z$. We find closed form solutions for both, $p(z)$ and $p(\mathbf{o},\mathbf{a}|z)$ given by

$$p^*(z) \propto \exp\big(\mathbb{E}_{p(\mathbf{o},\mathbf{a}|z)}[R_z(\mathbf{o},\mathbf{a})/\eta] + \mathcal{H}(\mathbf{o},\mathbf{a}|z)\big), \text{ and } \tilde{p}(\mathbf{o}_n,\mathbf{a}_n|z) = \exp\big(R_z(\mathbf{o}_n,\mathbf{a}_n)/\eta\big), \quad (5)$$

where $\tilde{p}(\mathbf{o}_n,\mathbf{a}_n|z)$ are the optimal unnormalized curricula, such that holds $p^*(\mathbf{o}_n,\mathbf{a}_n|z) = \tilde{p}(\mathbf{o}_n,\mathbf{a}_n|z)/\sum_n \tilde{p}(\mathbf{o}_n,\mathbf{a}_n|z)$. However, due to the hierarchical structure of $L(\boldsymbol{\psi},q)$ we implicitly optimize for $p(z)$ when updating the curricula. This result is frequently used throughout this work and formalized in Proposition 3.2. A proof can be found in Appendix A.2.

**Proposition 3.2.** *Let $p^*(z)$ and $\tilde{p}(\mathbf{o},\mathbf{a}|z)$ be the optimal mixture weights and unnormalized curricula for maximizing $L(\boldsymbol{\psi},q)$. It holds that*

$$p^*(z) = \sum_n \tilde{p}(\mathbf{o}_n,\mathbf{a}_n|z)/\sum_z\sum_n \tilde{p}(\mathbf{o}_n,\mathbf{a}_n|z).$$

The implicit updates of the mixture weights render the computation of $p^*(z)$ obsolete, reducing the optimization to computing the optimal (unnormalized) curricula $\tilde{p}(\mathbf{o},\mathbf{a}|z)$ and expert policy parameters $\boldsymbol{\theta}_z^*$. In particular, this result allows for training the policy using mini-batches and thus greatly improves the scalability to large datasets as explained in Section 3.5. Maximizing the lower bound with respect to $\boldsymbol{\theta}_z^*$ results in a weighted maximum likelihood estimation, i.e.,

$$\boldsymbol{\theta}_z^* = \arg\max_{\boldsymbol{\theta}_z} \sum_n \tilde{p}(\mathbf{o}_n,\mathbf{a}_n|z)\log p_{\boldsymbol{\theta}_z}(\mathbf{a}_n|\mathbf{o}_n,z), \quad (6)$$

where the curricula $\tilde{p}(\mathbf{o}_n,\mathbf{a}_n|z)$ assign sample weights. For further details on the M-step, including derivations of the closed-form solutions and the expert parameter objective see Appendix B.2.

### 3.4 Tightening the Lower Bound (E-Step)

Tightening of the lower bound (also referred to as E-step) is done by minimizing the expected Kullback-Leibler divergence in Equation 4. Using the properties of the KL divergence, it can easily be seen that the lower bound is tight if for all $n \in \{1,...,N\}$ $q(z|\mathbf{o}_n) = p(z|\mathbf{o}_n,\mathbf{a}_n)$ holds. To obtain $p(z|\mathbf{o}_n,\mathbf{a}_n)$ we leverage Bayes' rule, that is, $p(z|\mathbf{o}_n,\mathbf{a}_n) = p^*(z)p^*(\mathbf{o}_n,\mathbf{a}_n|z)/\sum_z p^*(z)p^*(\mathbf{o}_n,\mathbf{a}_n|z)$. Using Proposition 3.2 we find that

$$p(z|\mathbf{o}_n,\mathbf{a}_n) = \tilde{p}(\mathbf{o}_n,\mathbf{a}_n|z)/\sum_z \tilde{p}(\mathbf{o}_n,\mathbf{a}_n|z).$$

Please note that the lower bound is tight after every E-step as the KL divergence is set to zero. Thus, increasing the lower bound $L$ maximizes the original objective $J$ assuming that updates of $\boldsymbol{\theta}_z$ are not decreasing the expert policy log-likelihood $\log p_{\boldsymbol{\theta}_z}(\mathbf{a}|\mathbf{o},z)$.

### 3.5 Algorithmic Details

**Convergence Guarantees.** Proposition 3.3 establishes convergence guarantees for the mixture of experts policy objective $J$. The proof mainly relies on the facts that IMC has the same convergence guarantees as the EM algorithm and that Proposition 3.1 can be transferred to the per-component objective $J_z$. The full proof is given in Appendix A.1.

**Proposition 3.3.** *Let $J$ be defined as in Equation 4 and $0 < \eta < \infty$. Under mild assumptions on $p_{\boldsymbol{\theta}_z}$ and the optimization scheme for $\boldsymbol{\theta}_z$, it holds for all $i$, denoting the iteration index of the optimization process, that*

$$J(\boldsymbol{\psi}^{(i+1)}) \geq J(\boldsymbol{\psi}^{(i)}),$$

*where equality indicates that the IMC algorithm has converged to a local optimum.*

**Stopping Criterion.** We terminate the algorithm if either the maximum number of training iterations is reached or if the lower bound $L(\boldsymbol{\psi}, q)$ converges, i.e.,

$$|\Delta L| = |L^{(i)}(\boldsymbol{\psi}, q) - L^{(i-1)}(\boldsymbol{\psi}, q)| \leq \epsilon,$$

with threshold $\epsilon$ and two subsequent iterations $(i)$ and $(i-1)$. The lower bound can be evaluated efficiently using Corollary 3.2.1.

**Corollary 3.2.1.** *Consider the setup used in Proposition 3.2. For $p^*(z) \in \boldsymbol{\psi}$ and $\{p^*(\mathbf{o}, \mathbf{a}|z)\}_z \in \boldsymbol{\psi}$ it holds that*

$$L(\boldsymbol{\psi}, q) = \eta \log \sum_z \sum_n \tilde{p}(\mathbf{o}_n, \mathbf{a}_n|z).$$

See Appendix A.3 for a proof.

**Inference.** In order to perform inference, i.e., sample actions from the policy, we need to access the gating distribution for arbitrary observations $\mathbf{o} \in \mathcal{O}$ which is not possible as $p(z|\mathbf{o}, \mathbf{a})$ is only defined for observations contained in the dataset $\mathbf{o}, \mathbf{a}$. We therefore leverage Corollary 3.2.2 to learn an inference network $g_{\boldsymbol{\phi}}(z|\mathbf{o})$ with parameters $\boldsymbol{\phi}$ by minimizing the KL divergence between $p(z|\mathbf{o})$ and $g_{\boldsymbol{\phi}}(z|\mathbf{o})$ under $p(\mathbf{o})$ (see Appendix A.4 for a proof).

**Corollary 3.2.2.** *Consider the setup used in Proposition 3.2. It holds that*

$$\min_{\boldsymbol{\phi}} \mathbb{E}_{p(\mathbf{o})} D_{\mathrm{KL}}\big(p(z|\mathbf{o}) \| g_{\boldsymbol{\phi}}(z|\mathbf{o})\big) = \max_{\boldsymbol{\phi}} \sum_n \sum_z \tilde{p}(\mathbf{o}_n, \mathbf{a}_n|z) \log g_{\boldsymbol{\phi}}(z|\mathbf{o}_n).$$

Once trained, the inference network can be used for computing the exact log-likelihood as $p(\mathbf{a}|\mathbf{o}) = \sum_z g_{\boldsymbol{\phi}}(z|\mathbf{o}) p_{\boldsymbol{\theta}_z}(\mathbf{a}|\mathbf{o}, z)$ or sampling a component index, i.e., $z' \sim g_{\boldsymbol{\phi}}(z|\mathbf{o})$.

**Mini-Batch Updates.** Due to Proposition 3.2, the M- and E-step in the training procedure only rely on the unnormalized curricula $\tilde{p}(\mathbf{o}, \mathbf{a}|z)$. Consequently, there is no need to compute the normalization constant $\sum_n \tilde{p}(\mathbf{o}_n, \mathbf{a}_n|z)$. This allows us to utilize mini-batches instead of processing the entire dataset, resulting in an efficient scaling capability for large datasets. Please refer to Algorithm 1 for a detailed description of the complete training procedure.

## 4 Related Work

**Imitation Learning.** A variety of algorithms in imitation learning [1, 11] can be grouped into two categories: Inverse reinforcement learning [12, 13], which extracts a reward function from demonstrations and optimizes a policy subsequently, and behavioral cloning, which directly extracts a policy from demonstrations. Many works approach the problem of imitation learning by considering behavior cloning as a distribution-matching problem, in which the state distribution induced by the policy is required to align with the state distribution of the expert data. Some methods [14, 15] are based on adversarial methods inspired by Generative Adversarial Networks (GANs) [16]. A policy is trained to imitate the expert while a discriminator learns to distinguish between fake and expert data. However, these methods are not suitable for our case as they involve interacting with the environment during training. Other approaches focus on purely offline training and use various policy representations such as energy-based models [17], normalizing flows [18, 19], conditional variational autoencoders (CVAEs) [20, 21], transformers [22], or diffusion models [23, 24, 25, 26, 27, 28]. These

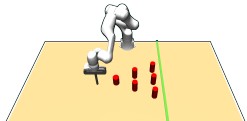 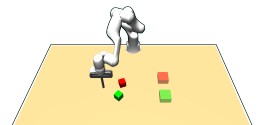 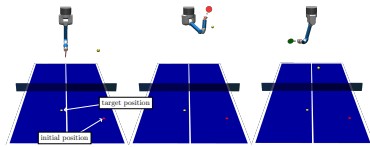

Figure 1: **Behavior learning environments:** Visualization of the obstacle avoidance task (left), the block pushing task (middle), and the table tennis task (right).

models can represent multi-modal expert distributions but are optimized based on the M-Projection, which leads to a performance decrease. Recent works [29, 30] have proposed training Mixture of Experts models with an objective similar to ours. However, the work by [29] requires environment-specific geometric features, which is not applicable in our setting, whereas the work by [30] considers linear experts and the learning of skills parameterized by motion primitives [31]. For a detailed differentiation between our work and the research conducted by [30], please refer to Appendix D.1.

**Curriculum Learning.** The authors of [32] introduced curriculum learning (CL) as a new paradigm for training machine learning models by gradually increasing the difficulty of samples that are exposed to the model. Several studies followed this definition [33, 34, 35, 36, 37, 38, 39]. Other studies used the term curriculum learning for gradually increasing the model complexity [40, 41, 42] or task complexity [43, 44, 45, 46]. All of these approaches assume that the difficulty-ranking of the samples is known a-priori. In contrast, we consider dynamically adapting the curriculum according to the learning progress of the model which is known as self-paced learning (SPL). Pioneering work in SPL was done in [47] which is related to our work in that the authors propose to update the curriculum as well as model parameters iteratively. However, their method is based on maximum likelihood which is different from our approach. Moreover, their algorithm is restricted to latent structural support vector machines. For a comprehensive survey on curriculum learning, the reader is referred to [34].

---

**Algorithm 1** IMC training procedure

1: **Require:** Data $\mathcal{D} = \{(\mathbf{o}_n, \mathbf{a}_n)\}_{n=1}^N$
2: **Require:** Number of components $N_z$
3: **Require:** Curriculum pacing $\eta$
4: **while** $|\Delta L| \leq \epsilon$ **do**
5:     Perform E-step:
6:     $q(z|\mathbf{o}_n, \mathbf{a}_n) \leftarrow \tilde{p}(\mathbf{o}_n, \mathbf{a}_n|z) / \sum_z \tilde{p}(\mathbf{o}_n, \mathbf{a}_n|z) \; \forall n$
7:     **for** $z \leftarrow 1, ..., N_z$ **do**
8:         Perform M-step for curricula:
9:         $R_z(\mathbf{o}_n, \mathbf{a}_n) \leftarrow \log p_{\boldsymbol{\theta}_z}(\mathbf{a}_n|\mathbf{o}_n, z) + \eta \log q(z|\mathbf{o}_n, \mathbf{a}_n) \; \forall n$
10:         $\tilde{p}(\mathbf{o}_n, \mathbf{a}_n|z) \leftarrow \exp\left(R_z(\mathbf{o}_n, \mathbf{a}_n)/\eta\right) \; \forall n$
11:         Perform M-step for experts:
12:         $\boldsymbol{\theta}_z^* \leftarrow \arg\max_{\boldsymbol{\theta}_z} \sum_n \tilde{p}(\mathbf{o}_n, \mathbf{a}_n|z) \log p_{\boldsymbol{\theta}_z}(\mathbf{a}_n|\mathbf{o}_n, z)$
13:     **end for**
14: **end while**

---

## 5 Experiments

We briefly outline the key aspects of our experimental setup.

**Experimental Setup.** For all experiments, we employ conditional Gaussian expert policies, i.e., $p_{\boldsymbol{\theta}_z}(\mathbf{a}|\mathbf{o}, z) = \mathcal{N}(\mathbf{a}|\boldsymbol{\mu}_{\boldsymbol{\theta}_z}(\mathbf{o}), \sigma^2 \mathbf{I})$. Please note that we parameterize the expert means $\boldsymbol{\mu}_{\boldsymbol{\theta}_z}$ using neural networks. For complex high-dimensional tasks, we share features between different expert policies by introducing a deep neural network backbone. Moreover, we use a fixed variance of $\sigma^2 = 1$ and $N_z = 50$ components for all experiments and tune the curriculum pacing $\eta$. For more details see Appendix C.2.

**Baselines.** We compare our method to state-of-the-art generative models including denoising diffusion probabilistic models (DDPM) [24], normalizing flows (NF) [18] and conditional variational autoencoders (CVAE) [20]. Moreover, we consider energy-based models for behavior learning (IBC) [17] and the recently proposed behavior transformer (BeT) [22]. Lastly, we compare against mixture of experts trained using expectation maximization (EM) [48] and backpropagation (MDN) [8] and the ML-Cur algorithm [30]. We extensively tune the hyperparameters of the baselines using Bayesian optimization [49] on all experiments.

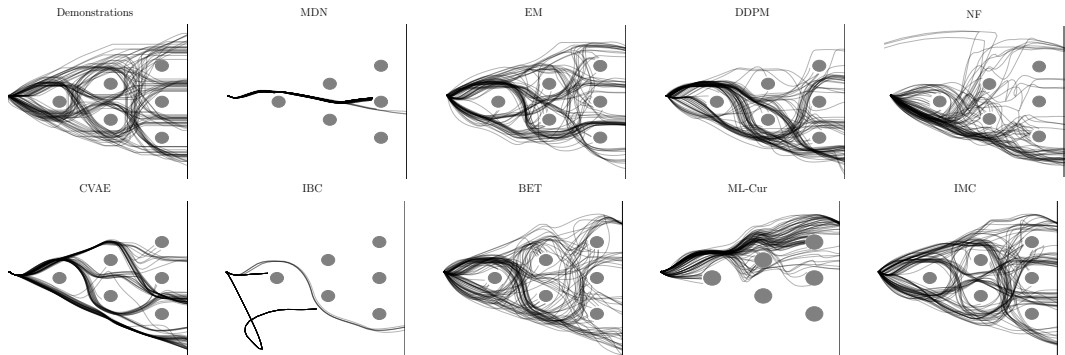

Figure 2: **Obstacle Avoidance:** Visualization of 100 end-effector trajectories for all trained models. Every method is trained on diverse demonstration data (left). MDN, NF, and IBC mostly fail to successfully solve the task. Other methods (EM, DDPM, CVAE, BET, ML-Cur) either fail to have high success rates or disregard modes in the data. Only IMC is able to perform well on both metrics.

Table 1: **Performance Table**: We compare IMC against strong baselines on three complex robot manipulation tasks. The best results use bold formatting. $\uparrow$ indicates that higher and $\downarrow$ that lower values are better. For further details, please refer to the accompanying text.

| | OBSTACLE AVOIDANCE | | BLOCK PUSHING | | | TABLE TENNIS | |
| | SUCCESS RATE ($\uparrow$) | ENTROPY ($\uparrow$) | SUCCESS RATE ($\uparrow$) | ENTROPY ($\uparrow$) | DISTANCE ERROR ($\downarrow$) | SUCCESS RATE ($\uparrow$) | DISTANCE ERROR ($\downarrow$) |
|---|---|---|---|---|---|---|---|
| MDN | $0.200_{\pm0.421}$ | $0.000_{\pm0.000}$ | $0.000_{\pm0.000}$ | $0.000_{\pm0.000}$ | $0.360_{\pm0.005}$ | $0.031_{\pm0.013}$ | $0.549_{\pm0.056}$ |
| EM | $0.675_{\pm0.033}$ | $0.902_{\pm0.035}$ | $0.458_{\pm0.048}$ | $0.707_{\pm0.042}$ | $0.127_{\pm0.011}$ | $0.725_{\pm0.042}$ | $0.220_{\pm0.012}$ |
| DDPM | $0.719_{\pm0.075}$ | $0.638_{\pm0.079}$ | $0.516_{\pm0.036}$ | $0.713_{\pm0.043}$ | $0.123_{\pm0.007}$ | $0.866_{\pm0.010}$ | $0.185_{\pm0.007}$ |
| NF | $0.313_{\pm0.245}$ | $0.349_{\pm0.208}$ | $0.001_{\pm0.001}$ | $0.000_{\pm0.000}$ | $0.346_{\pm0.034}$ | $0.422_{\pm0.035}$ | $0.371_{\pm0.013}$ |
| CVAE | $0.853_{\pm0.113}$ | $0.465_{\pm0.183}$ | $0.505_{\pm0.089}$ | $0.162_{\pm0.071}$ | $0.123_{\pm0.027}$ | $0.620_{\pm0.050}$ | $0.320_{\pm0.010}$ |
| IBC | $0.379_{\pm0.411}$ | $0.098_{\pm0.131}$ | $0.482_{\pm0.026}$ | $\mathbf{0.758_{\pm0.022}}$ | $0.123_{\pm0.007}$ | $0.567_{\pm0.030}$ | $0.310_{\pm0.010}$ |
| BET | $0.504_{\pm0.076}$ | $0.837_{\pm0.066}$ | $0.374_{\pm0.041}$ | $0.607_{\pm0.037}$ | $0.151_{\pm0.008}$ | $0.758_{\pm0.025}$ | $0.235_{\pm0.011}$ |
| ML-CUR | $0.454_{\pm0.223}$ | $0.035_{\pm0.024}$ | $0.000_{\pm0.000}$ | $0.000_{\pm0.000}$ | $0.408_{\pm0.030}$ | $0.836_{\pm0.020}$ | $0.181_{\pm0.011}$ |
| IMC | $\mathbf{0.855_{\pm0.053}}$ | $\mathbf{0.930_{\pm0.031}}$ | $\mathbf{0.521_{\pm0.045}}$ | $0.654_{\pm0.041}$ | $\mathbf{0.120_{\pm0.014}}$ | $\mathbf{0.870_{\pm0.017}}$ | $\mathbf{0.153_{\pm0.007}}$ |

**Evaluation.** For all experiments we perform multiple simulations using the trained policies to compute the performance metrics. Firstly, we report the *success rate*, which is the fraction of simulations that led to successful task completion, reflecting the susceptibility to mode averaging. Secondly, we compute a task-specific (categorical) distribution over pre-defined behaviors that lead to successful task completion. Thereafter, we compute the *entropy* over this distribution to quantify the diversity in the learned behaviors and therefore the ability to cover multiple modes present in the data distribution. An entropy of 0 implies that a model executes the same behavior, while higher entropy values indicate that the model executes different behaviors. Further details are provided in the descriptions of individual tasks.

We report the mean and the standard deviation over ten random seeds for all experiments. For a detailed explanation of tasks, data, performance metrics, and hyperparameters see Appendix C. For further experimental results see Appendix E. The code is available online [2].

## 5.1 Obstacle Avoidance

The obstacle avoidance environment is visualized in Figure 1 (left) and consists of a seven DoF Franka Emika Panda robot arm equipped with a cylindrical end effector simulated using the MuJoCo physics engine [50]. The goal of the task is to reach the green finish line without colliding with one of the six obstacles. The dataset contains four human demonstrations for all ways of avoiding obstacles and completing the task. It is collected using a game-pad controller and inverse kinematics (IK) in the xy-plane amounting to 7.3k $(\mathbf{o}, \mathbf{a})$ pairs. The observations $\mathbf{o} \in \mathbb{R}^4$ contain the end-effector position and velocity of the robot. The actions $\mathbf{a} \in \mathbb{R}^2$ represent the desired position of the robot. There are 24 different ways of avoiding obstacles and reaching the green finish line, each of which we define as a different behavior $\beta$. At test time, we perform 1000 simulations for computing the success rate and the entropy $\mathcal{H}(\beta) = -\sum_{\beta} p(\beta) \log_{24} p(\beta)$. Please note that we use $\log_{24}$ for the purpose of

---

[2]https://github.com/ALRhub/imc

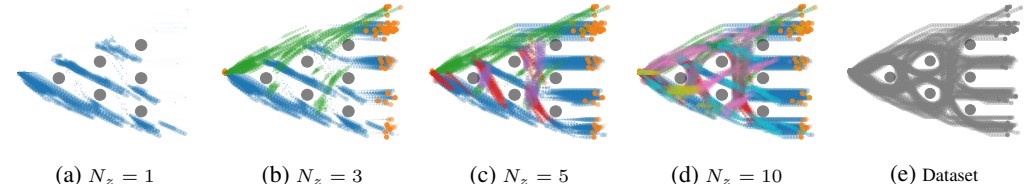

(a) $N_z = 1$    (b) $N_z = 3$    (c) $N_z = 5$    (d) $N_z = 10$    (e) Dataset

Figure 3: **Curriculum Visualization:** Visualization of the curricula $p(\mathbf{o}, \mathbf{a}|z)$ for a different number of components $N_z$ on the obstacle avoidance task. The color indicates different components $z$ and the size of the dots is proportional to $p(\mathbf{o}_n, \mathbf{a}_n|z)$. For $N_z = 1$ we observe that the model ignores most samples in the dataset as a single expert is not able to achieve high log-likelihood values $p_{\boldsymbol{\theta}_z}$ on all samples (Figure 3a). Adding more components to the model results in higher coverage of samples as shown in Figure 3b-3d. Figure 3e visualizes all samples contained in the dataset.

enhancing interpretability, as it ensures $\mathcal{H}(\beta) \in [0, 1]$. An entropy value of 0 signifies a policy that consistently executes the same behavior, while an entropy value of 1 represents a diverse policy that executes all behaviors with equal probability and hence matches the true behavior distribution by design of the data collection process. The results are shown in Table 1. Additionally, we provide a visualization of the learned curriculum in Figure 3. Further details are provided in Appendix C.1.1.

## 5.2 Block Pushing

The block pushing environment is visualized in Figure 1 (middle) and uses the setup explained in Section 5.1. The robot manipulator is tasked to push blocks into target zones. Having two blocks and target zones amounts to four different push sequences (see e.g. Figure 8), each of which we define as a different behavior $\beta$. Using a gamepad, we recorded $500$ demonstrations for each of the four push sequences with randomly sampled initial block configurations $\mathbf{o}_0$ (i.e., initial positions and orientations), amounting to a total of $463k$ $(\mathbf{o}, \mathbf{a})$ pairs. The observations $\mathbf{o} \in \mathbb{R}^{16}$ contain information about the robot's state and the block configurations. The actions $\mathbf{a} \in \mathbb{R}^2$ represent the desired position of the robot. We evaluate the models using three different metrics: Firstly, the *success rate* which is the proportion of trajectories that manage to push both boxes to the target zones. Secondly, the *entropy*, which is computed over different push sequences $\beta$. Since high entropy values can be achieved by following different behaviors for different initial block configurations $\mathbf{o}_0$ in a deterministic fashion, the entropy of $p(\beta)$ can be a poor metric for quantifying diversity. Hence, we evaluate the expected entropy conditioned on the initial state $\mathbf{o}_0$, i.e., $\mathbb{E}_{p(\mathbf{o}_0)}\left[\mathcal{H}(\beta|\mathbf{o}_0)\right] \approx -\frac{1}{N_0} \sum_{\mathbf{o}_0 \sim p(\mathbf{o}_0)} \sum_\beta p(\beta|\mathbf{o}_0) \log_4 p(\beta|\mathbf{o}_0)$. If, for the same $\mathbf{o}_0$, all behaviors can be achieved, the expected entropy is high. In contrast, the entropy is 0 if the same behavior is executed for the same $\mathbf{o}_0$. Here, $p(\mathbf{o}_0)$ and $N_0$ denote the distribution over initial block configurations and the number of samples respectively. See Appendix C.1.2 for more details. Lastly, we quantify the performance on non-successful trajectories, via *distance error*, i.e., the distance from the blocks to the target zones at the end of a trajectory. The success rate and distance error indicate whether a model is able to avoid averaging over different behaviors. The entropy assesses the ability to represent multimodal data distributions by completing different push sequences. The results are reported in Table 1 and generated by simulating 16 evaluation trajectories for 30 different initial block configurations $\mathbf{o}_0$ per seed. The difficulty of the task is reflected by the low success rates of most models. Besides being a challenging manipulation task, the high task complexity is caused by having various sources of multimodality in the data distribution: First, the inherent versatility in human behavior. Second, multiple human demonstrators, and lastly different push sequences for the same block configuration.

## 5.3 Franka Kitchen

The Franka kitchen environment was introduced in [51] and uses a seven DoF Franka Emika Panda robot with a two DoF gripper to interact with a simulated kitchen environment. The corresponding dataset contains 566 human-collected trajectories recorded using a virtual reality setup amounting to 128k $(\mathbf{o}, \mathbf{a})$ pairs. Each trajectory executes a sequence completing four out of seven different tasks. The observations $\mathbf{o} \in \mathbb{R}^{30}$ contain information about the position and orientation of the task-relevant objects in the environment. The actions $\mathbf{a} \in \mathbb{R}^9$ represent the control signals for

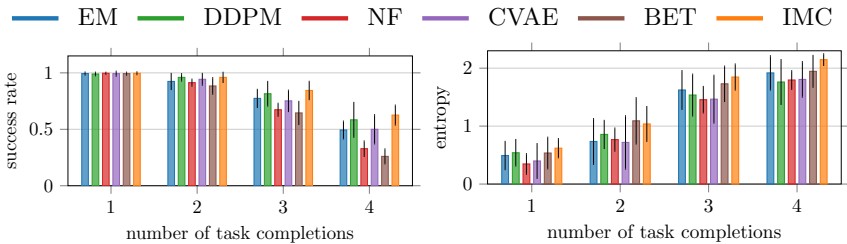

Figure 4: **Franka Kitchen:** Performance comparison between various policy learning algorithms, evaluating the success rate and entropy for a varying number of task completions. IMC is able to outperform the baselines in both metrics, achieving a high success rate while showing a higher diversity in the task sequences. Performance comparison between various policy learning algorithms. IMC is more successful in completing tasks (success rate) while at the same time having the highest diversity in the sequence of task completions (entropy).

the robot and gripper. To assess a model's ability to avoid mode averaging we again use the *success rate* over the number of tasks solved within one trajectory. For each number of solved tasks $\in \{1, 2, 3, 4\}$, we define a behavior $\beta$ as the order in which the task is completed and use the entropy $\mathcal{H}(\beta) = -\sum_{\beta} p(\beta) \log p(\beta)$ to quantify diversity. The results are shown in Figure 5 and are generated using 100 evaluation trajectories for each seed. There are no results reported for IBC, MDN and ML-Cur as we did not manage to obtain reasonable results.

## 5.4 Table Tennis

The table tennis environment is visualized in Figure 1 (right) and consists of a seven DOF robot arm equipped with a table tennis racket and is simulated using the MuJoCo physics engine. The goal is to return the ball to varying target positions after it is launched from a randomized initial position. Although not collected by human experts, the 5000 demonstrations are generated using a reinforcement learning (RL) agent that is optimized for highly multimodal behavior such as backhand and forehand strokes [52]. Each demonstration consists of an observation $\mathbf{o} \in \mathbb{R}^4$ defining the initial and target ball position. Movement primitives (MPs) [31] are used to describe the joint space trajectories of the robot manipulator using two basis functions per joint and thus $\mathbf{a} \in \mathbb{R}^{14}$. We evaluate the model performance using the *success rate*, that is, how frequently the ball is returned to the other side. Moreover, we employ the *distance error*, i.e., the Euclidean distance from the landing position of the ball to the target position. Both metrics reflect if a model is able to avoid averaging over different movements. For this experiment, we do not report the entropy as we do not know the various behaviors executed by the RL agent. The results are shown in Table 1 and are generated using 500 different initial and target positions. Note that the reinforcement learning agent used to generate the data achieves an average success rate of 0.91 and a distance error of 0.14 which is closely followed by IMC.

## 5.5 Ablation Studies

Additionally, we compare the performance of IMC with EM for a varying number of components on the obstacle avoidance and table tennis task. The results are shown in Figure 4 and highlight the properties of the moment and information projection: Using limited model complexity, e.g. 1 or 5 components, EM suffers from mode averaging, resulting in poor performances (Figure 5a and Figure 5b). This is further illustrated in Figure 5c. In contrast, the zero forcing property of the information projection allows IMC to avoid mode averaging (see Figure 5d) which is reflected in the success rates and distance error for a small number of components. The performance gap between EM and IMC for high model complexities suggests that EM still suffers from averaging problems. Moreover, the results show that IMC needs fewer components to achieve the same performance as EM.

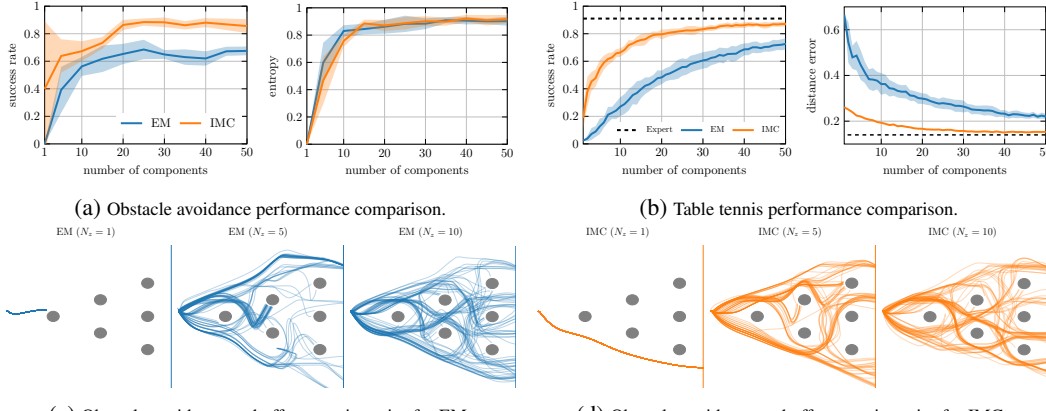

(a) Obstacle avoidance performance comparison.

(b) Table tennis performance comparison.

(c) Obstacle avoidance end-effector trajectories for EM.

(d) Obstacle avoidance end-effector trajectories for IMC.

Figure 5: **Obstacle Avoidance:** IMC and EM improve the success rate and entropy with an increasing number of components (a). For a small number of components, IMC archives a high success rate, as it allows the policy to focus on data that it can represent. In contrast, the policy trained with EM fails as it is forced to cover the whole data. This is visualized in the end-effector trajectories (c + d). Similar observations can be made for **Table Tennis:** Both performance metrics increase with a higher number of components. IMC manages to achieve good performance with a small number of components.

# 6 Conclusion

We presented *Information Maximizing Curriculum* (IMC), a novel approach for conditional density estimation, specifically designed to address mode-averaging issues commonly encountered when using maximum likelihood-based optimization in the context of multimodal density estimation. IMC's focus on mitigating mode-averaging is particularly relevant in imitation learning from human demonstrations, where the data distribution is often highly multimodal due to the diverse and versatile nature of human behavior.

IMC uses a curriculum to assign weights to the training data allowing the policy to focus on samples it can represent, effectively mitigating the mode-averaging problem. We extended our approach to a mixture of experts (MoE) policy, where each mixture component selects its own subset of the training data for learning, allowing the model to imitate the rich and versatile behavior present in the demonstration data.

Our experimental results demonstrate the superior performance of our method compared to state-of-the-art policy learning algorithms and mixture of experts (MoE) policies trained using competing optimization algorithms. Specifically, on complex multimodal simulated control tasks with data collected from human demonstrators, our method exhibits the ability to effectively address two key challenges: *i)* avoiding mode averaging and *ii)* covering all modes present in the data distribution.

**Limitations.** While our current approach achieves state-of-the-art performance, there are still areas for improvement in parameterizing our model. Presently, we employ simple multilayer perceptrons to parameterize the expert policies. However, incorporating image observations would require a convolutional neural network (CNN) [53] backbone. Additionally, our current model relies on the Markov assumption, but relaxing this assumption and adopting history-based models like transformers [54] could potentially yield significant performance improvements. Lastly, although this work primarily concentrates on continuous domains, an intriguing prospect for future research would be to explore the application of IMC in discrete domains.

**Broader Impact.** Improving imitation learning algorithms holds the potential to enhance the accessibility of robotic systems in real-world applications, with both positive and negative implications. We acknowledge that identifying and addressing any potential adverse effects resulting from the deployment of these robotic systems is a crucial responsibility that falls on sovereign governments.

## Acknowledgments and Disclosure of Funding

This work was supported by funding from the pilot program Core Informatics of the Helmholtz Association (HGF). The authors acknowledge support by the state of Baden-Württemberg through bwHPC, as well as the HoreKa supercomputer funded by the Ministry of Science, Research and the Arts Baden-Württemberg and by the German Federal Ministry of Education and Research.

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

# A Proofs

## A.1 Proof of Proposition 3.1 and 3.3

**Convergence of the Single Expert Objective (Proposition 3.1).** We perform coordinate ascent on $\tilde{J}$ which is guaranteed to converge to a stationary point if updating each coordinate results in a monotonic improvement of [9]. For fixed expert parameters we find the unique $p(\mathbf{o}, \mathbf{a})$ that maximizes $\tilde{J}$ [55] (see Section 3.1) and hence we have $\tilde{J}(p(\mathbf{o}, \mathbf{a})^{(i)}, \boldsymbol{\theta}) \geq \tilde{J}(p(\mathbf{o}, \mathbf{a})^{(i-1)}, \boldsymbol{\theta})$ where $i$ denotes the iteration. Under suitable assumptions ($\log p_{\boldsymbol{\theta}}$ is differentiable, its gradient is $L$-Lipschitz, $\boldsymbol{\theta}_z$ is updated using gradient ascent and the learning rate is chosen such that the descent lemma [56] holds), it holds that $\tilde{J}(p(\mathbf{o}, \mathbf{a}), \boldsymbol{\theta}^{(i)}) \geq \tilde{J}(p(\mathbf{o}, \mathbf{a}), \boldsymbol{\theta}^{(i-1)})$. Hence, we are guaranteed to converge to a stationary point of $\tilde{J}$.    $\square$

**Convergence of the Mixture of Experts Objective (Proposition 3.3).** As we tighten the lower bound $L$ in every E-step, it remains to show that $L(\boldsymbol{\psi}^{(i)}, q) \geq L(\boldsymbol{\psi}^{(i-1)}, q)$ to prove that $J(\boldsymbol{\psi})$ is guaranteed to converge to a stationary point, with $\boldsymbol{\psi} = \{p(z), \{p(\mathbf{o}, \mathbf{a}|z)\}_z, \{\boldsymbol{\theta}_z\}_z\}$. To that end, we again perform a coordinate ascent on $L$ to show a monotonic improvement in every coordinate. Note that we find the unique $p(z)$ and $\{p(\mathbf{o}, \mathbf{a}|z)\}_z$ that maximize $L$ via Proposition 3.2 and Equation 5. Analogously to the proof of Proposition 3.1 we can show monotonic improvement of $L$ in $\{\boldsymbol{\theta}_z\}_z$ under suitable assumptions on $\log p_{\boldsymbol{\theta}_z}$.    $\square$

**Remark.** Although *stochastic* gradient ascent does not guarantee strictly monotonic improvements in the objective function $J$, our empirical observations suggest that $J$ indeed tends to increase monotonically in practice as shown by Figure 6.

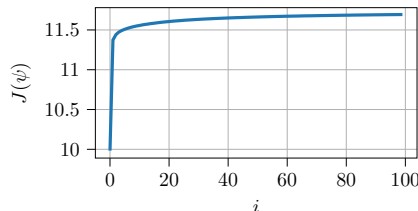

Figure 6: **Convergence of the Mixture of Experts Objective.** Objective function value $J$ for 100 training iterations $i$ on Franka Kitchen.

## A.2 Proof of Proposition 3.2

Expanding the entropy in Equation 5 we obtain

$$p^*(z) \propto \exp\left(\mathbb{E}_{p^*(\mathbf{o}, \mathbf{a}|z)}[R_z(\mathbf{o}, \mathbf{a})/\eta - \log p^*(\mathbf{o}, \mathbf{a}|z)]\right).$$

Using $p^*(\mathbf{o}, \mathbf{a}|z) = \tilde{p}(\mathbf{o}, \mathbf{a}|z)/\sum_n \tilde{p}(\mathbf{o}_n, \mathbf{a}_n|z)$ yields

$$p^*(z) \propto \exp\left(\mathbb{E}_{p^*(\mathbf{o}, \mathbf{a}|z)}[R_z(\mathbf{o}, \mathbf{a})/\eta - \log \tilde{p}(\mathbf{o}, \mathbf{a}|z) + \log \sum_n \tilde{p}(\mathbf{o}_n, \mathbf{a}_n|z)]\right).$$

Next, leveraging that $\log \tilde{p}(\mathbf{o}, \mathbf{a}|z) = R_z(\mathbf{o}_n, \mathbf{a}_n)/\eta$ we see that

$$p^*(z) \propto \exp\left(\mathbb{E}_{p^*(\mathbf{o}, \mathbf{a}|z)}[\log \sum_n \tilde{p}(\mathbf{o}_n, \mathbf{a}_n|z)]\right) = \sum_n \tilde{p}(\mathbf{o}_n, \mathbf{a}_n|z),$$

which concludes the proof.    $\square$

## A.3 Proof of Corollary 3.2.1

We start by rewriting the lower bound as $L(\boldsymbol{\psi}, q) =$

$$\mathbb{E}_{p^*(z)}\left[\mathbb{E}_{p^*(\mathbf{o}, \mathbf{a}|z)}[R_z(\mathbf{o}, \mathbf{a}) - \eta \log p^*(\mathbf{o}, \mathbf{a}|z)] - \eta \log p^*(z)\right].$$

Using $p^*(\mathbf{o}, \mathbf{a}|z) \propto \tilde{p}(\mathbf{o}, \mathbf{a}|z)$ and Proposition 3.2 we obtain

$$L(\boldsymbol{\psi}, q) = \mathbb{E}_{p^*(z)}\Big[\mathbb{E}_{p^*(\mathbf{o},\mathbf{a}|z)}[R_z(\mathbf{o}, \mathbf{a}) - \eta \log \tilde{p}(\mathbf{o}, \mathbf{a}|z)$$
$$+ \eta \log \sum_n \tilde{p}(\mathbf{o}_n, \mathbf{a}_n|z)] - \eta \log \sum_n \tilde{p}(\mathbf{o}_n, \mathbf{a}_n|z) + \eta \log \sum_z \sum_n \tilde{p}(\mathbf{o}_n, \mathbf{a}_n|z)\Big]$$

With $\eta \log \tilde{p}(\mathbf{o}, \mathbf{a}|z) = R_z(\mathbf{o}_n, \mathbf{a}_n)$ all most terms cancel, giving

$$L(\boldsymbol{\psi}, q) = \mathbb{E}_{p^*(z)}\Big[\eta \log \sum_z \sum_n \tilde{p}(\mathbf{o}_n, \mathbf{a}_n|z)\Big] = \eta \log \sum_z \sum_n \tilde{p}(\mathbf{o}_n, \mathbf{a}_n|z),$$

which concludes the proof. $\quad\square$

### A.4   Proof of Corollary 3.2.2

$$\min_{\phi} \mathbb{E}_{p(\mathbf{o})} D_{\mathrm{KL}}(p(z|\mathbf{o})\|g_\phi(z|\mathbf{o})) = \max_{\phi} \int_{\mathcal{O}} \sum_z p(\mathbf{o}, z) \log g_\phi(z|\mathbf{o}) \mathrm{d}\mathbf{o}$$

$$= \max_{\phi} \int_{\mathcal{O}} \int_{\mathcal{A}} \sum_z p(\mathbf{o}, \mathbf{a}, z) \log g_\phi(z|\mathbf{o}) \mathrm{d}\mathbf{a}\mathrm{d}\mathbf{o} = \max_{\phi} \int_{\mathcal{O}} \int_{\mathcal{A}} p(\mathbf{o}, \mathbf{a}) \sum_z p(z|\mathbf{o}, \mathbf{a}) \log g_\phi(z|\mathbf{o}) \mathrm{d}\mathbf{a}\mathrm{d}\mathbf{o}$$

$$= \min_{\phi} \mathbb{E}_{p(\mathbf{o},\mathbf{a})} D_{\mathrm{KL}}(p(z|\mathbf{o}, \mathbf{a})\|g_\phi(z|\mathbf{o})).$$

Expanding the expected KL divergence, we get

$$\min_{\phi} \mathbb{E}_{p(\mathbf{o},\mathbf{a})} D_{\mathrm{KL}}\big(p(z|\mathbf{o}, \mathbf{a})\|g_\phi(z|\mathbf{o})\big) = \min_{\phi} \sum_n p(\mathbf{o}_n, \mathbf{a}_n) \sum_z p(z|\mathbf{o}_n, \mathbf{a}_n) \log \frac{p(z|\mathbf{o}_n, \mathbf{a}_n)}{g_\phi(z|\mathbf{o}_n)}.$$

Noting that $p(z|\mathbf{o}_n, \mathbf{a}_n)$ is independent of $\phi$ we can rewrite the objective as

$$\max_{\phi} \sum_n p(\mathbf{o}_n, \mathbf{a}_n) \sum_z p(z|\mathbf{o}_n, \mathbf{a}_n) \log g_\phi(z|\mathbf{o}_n).$$

Using that $p(z|\mathbf{o}, \mathbf{a}) = \tilde{p}(\mathbf{o}, \mathbf{a}|z)/\sum_z \tilde{p}(\mathbf{o}, \mathbf{a}|z)$ together with $p(\mathbf{o}, \mathbf{a}) = \sum_z p^*(z)p^*(\mathbf{o}, \mathbf{a}|z)$ yields

$$\max_{\phi} \sum_n \sum_z p^*(z)p^*(\mathbf{o}_n, \mathbf{a}_n|z) \sum_z \frac{\tilde{p}(\mathbf{o}_n, \mathbf{a}_n|z)}{\sum_z \tilde{p}(\mathbf{o}_n, \mathbf{a}_n|z)} \log g_\phi(z|\mathbf{o}_n).$$

Using Proposition 3.2 we can rewrite $p^*(z)p^*(\mathbf{o}, \mathbf{a}|z)$ as $\tilde{p}(\mathbf{o}, \mathbf{a}|z)/\sum_z \sum_n \tilde{p}(\mathbf{o}_n, \mathbf{a}_n|z)$. Since the constant factor $1/\sum_z \sum_n \tilde{p}(\mathbf{o}_n, \mathbf{a}_n|z)$ does not affect the optimal value of $\phi$ we obtain

$$\max_{\phi} \sum_n \sum_z \tilde{p}(\mathbf{o}_n, \mathbf{a}_n|z) \sum_z \frac{\tilde{p}(\mathbf{o}_n, \mathbf{a}_n|z)}{\sum_z \tilde{p}(\mathbf{o}_n, \mathbf{a}_n|z)} \log g_\phi(z|\mathbf{o}_n) = \max_{\phi} \sum_n \sum_z \tilde{p}(\mathbf{o}_n, \mathbf{a}_n|z) \log g_\phi(z|\mathbf{o}_n),$$

which concludes the proof. $\quad\square$

## B   Derivations

### B.1   Lower Bound Decomposition

To arrive at Equation 4 by marginalizing over the latent variable $o$ for the entropy of the joint curriculum, i.e.,

$$\mathcal{H}(\mathbf{o}, \mathbf{a}) = -\sum_n p(\mathbf{o}_n, \mathbf{a}_n) \log p(\mathbf{o}_n, \mathbf{a}_n)$$

$$= -\sum_n p(\mathbf{o}_n, \mathbf{a}_n) \sum_z p(z|\mathbf{o}_n, \mathbf{a}_n) \log p(\mathbf{o}_n, \mathbf{a}_n)$$

Next, we use Bayes' theorem, that is, $p(\mathbf{o}_n, \mathbf{a}_n) = p(z)p(\mathbf{o}_n, \mathbf{a}_n|z)/p(z|\mathbf{o}_n, \mathbf{a}_n)$, giving

$$\mathcal{H}(\mathbf{o}, \mathbf{a}) = -\sum_n p(\mathbf{o}_n, \mathbf{a}_n) \sum_z p(z|\mathbf{o}_n, \mathbf{a}_n)\big(\log p(z) + \log p(\mathbf{o}_n, \mathbf{a}_n|z) - \log p(z|\mathbf{o}_n, \mathbf{a}_n)\big).$$

Moreover, we add and subtract the log auxiliary distribution $\log q(z|\mathbf{o}_n, \mathbf{a}_n)$ which yields

$$\mathcal{H}(\mathbf{o}, \mathbf{a}) = -\sum_n p(\mathbf{o}_n, \mathbf{a}_n) \sum_z p(z|\mathbf{o}_n, \mathbf{a}_n)\big(\log p(z) + \log p(\mathbf{o}_n, \mathbf{a}_n|z)$$
$$-\log p(z|\mathbf{o}_n, \mathbf{a}_n) + \log q(z|\mathbf{o}_n, \mathbf{a}_n) - \log q(z|\mathbf{o}_n, \mathbf{a}_n)\big).$$

Rearranging the terms leads and writing the sums in terms of expectations we arrive at

$$\mathcal{H}(\mathbf{o}, \mathbf{a}) = \mathbb{E}_{p(z)}\big[\mathbb{E}_{p(z|\mathbf{o},\mathbf{a})}[\log q(z|\mathbf{o},\mathbf{a})] + \mathcal{H}(\mathbf{o},\mathbf{a}|z)\big] + \mathcal{H}(z) + D_{\text{KL}}\big(p(z|\mathbf{o},\mathbf{a})\|q(z|\mathbf{o},\mathbf{a})\big).$$

Lastly, multiplying $\mathcal{H}(\mathbf{o}, \mathbf{a})$ with $\eta$ and adding $\mathbb{E}_{p(z)}\mathbb{E}_{p(\mathbf{o},\mathbf{a}|z)}[\log p_{\boldsymbol{\theta}_z}(\mathbf{y}|\mathbf{x}, z)]$ we arrive at Equation 4 which concludes the derivation.

## B.2  M-Step Objectives

**Closed-Form Curriculum Updates.** In order to derive the closed-form solution to Equation 5 (RHS) we solve

$$\max_{p(\mathbf{o},\mathbf{a}|z)} J_z(p(\mathbf{o}, \mathbf{a}|z), \boldsymbol{\theta}_z) = \max_{p(\mathbf{o},\mathbf{a}|z)} \mathbb{E}_{p(\mathbf{o},\mathbf{a}|z)}[R_z(\mathbf{o}, \mathbf{a})] + \eta\mathcal{H}(\mathbf{o}, \mathbf{a}|z) \quad \text{subject to} \quad \sum_n p(\mathbf{o}, \mathbf{a}) = 1.$$

Following the procedure of constrained optimization, we write down the Lagrangian function [9] as

$$\mathcal{L}(p, \lambda) = \sum_n p(\mathbf{o}_n, \mathbf{a}_n|z)R_z(\mathbf{o}_n, \mathbf{a}_n) - \eta\sum_n p(\mathbf{o}_n, \mathbf{a}_n|z)\log p(\mathbf{o}_n, \mathbf{a}_n|z) + \lambda(\sum_n p(\mathbf{o}_n, \mathbf{a}_n|z) - 1),$$

where $\lambda$ is the Lagrangian multiplier. As $p$ is discrete, we solve for the optimal entries of $p(\mathbf{o}_n, \mathbf{a}_n|z)$, that is, $p'(\mathbf{o}_n, \mathbf{a}_n, \lambda|z) = \arg\max_p \mathcal{L}(p, \lambda)$. Setting the partial derivative of $\mathcal{L}(p, \lambda)$ with respect to $p$ zero, i.e.,

$$\frac{\partial}{\partial p(\mathbf{o}_n, \mathbf{a}_n|z)}\mathcal{L}(p, \lambda) = R_z(\mathbf{o}_n, \mathbf{a}_n) - \eta\log p(\mathbf{o}_n, \mathbf{a}_n|z) - \eta + \lambda \overset{!}{=} 0.$$

yields $p'(\mathbf{o}_n, \mathbf{a}_n, \lambda|z) = \exp\big(R_z(\mathbf{o}_n, \mathbf{a}_n) - \eta + \lambda\big)/\eta$.

Plugging $p'$ back in the Lagrangian gives the dual function $g(\lambda)$, that is,

$$g(\eta) = \mathcal{L}(p', \lambda) = -\eta + \eta\sum_n \exp\big(R_z(\mathbf{o}_n, \mathbf{a}_n) - \eta + \lambda\big)/\eta.$$

Solving for $\lambda^* = \arg\min_{\lambda\geq 0} g(\lambda)$ equates to

$$\frac{\partial}{\partial\lambda}g(\lambda) = -1 + \eta\sum_n \exp\big(R_z(\mathbf{o}_n, \mathbf{a}_n) - \eta + \lambda\big)/\eta \overset{!}{=} 0$$

$$\iff \lambda^* = -\log\Big(\eta\sum_n \exp\big(R_z(\mathbf{o}_n, \mathbf{a}_n) - \eta\big)/\eta\Big).$$

Finally, substituting $\lambda^*$ into $p'$ we have

$$p^*(\mathbf{o}_n, \mathbf{a}_n|z) = p'(\mathbf{o}_n, \mathbf{a}_n, \lambda^*|z) = \frac{\exp\big(R_z(\mathbf{o}_n, \mathbf{a}_n)/\eta\big)}{\sum_n \exp\big(R_z(\mathbf{o}_n, \mathbf{a}_n)/\eta\big)},$$

which concludes the derivation. The derivation of the optimal mixture weights $p^*(z)$ works analogously.

**Expert Objective.** In order to derive the expert objective of Equation 6 we solve $\max_{\boldsymbol{\theta}_z} J_z(p(\mathbf{o}, \mathbf{a}|z), \boldsymbol{\theta}_z) =$

$$\max_{\boldsymbol{\theta}_z} \sum_n p(\mathbf{o}_n, \mathbf{a}_n|z)\Big(\log p_{\boldsymbol{\theta}_z}(\mathbf{a}_n|\mathbf{o}_n, z) + \eta\log q(z|\mathbf{o}_n, \mathbf{a}_n) - \eta\log p(\mathbf{o}_n, \mathbf{a}_n|z)\Big).$$

Noting that $q(z|\mathbf{o}_n, \mathbf{a}_n)$ and $p(\mathbf{o}_n, \mathbf{a}_n|z)$ are independent of $\boldsymbol{\theta}_z$ and $p(\mathbf{o}_n, \mathbf{a}_n|z) = \tilde{p}(\mathbf{o}_n, \mathbf{a}_n|z)/\sum_n \tilde{p}(\mathbf{o}_n, \mathbf{a}_n|z)$ we find that

$$\max_{\boldsymbol{\theta}_z} J_z(p(\mathbf{o}, \mathbf{a}|z), \boldsymbol{\theta}_z) = \max_{\boldsymbol{\theta}_z} \sum_n \frac{\tilde{p}(\mathbf{o}_n, \mathbf{a}_n|z)}{\sum_n \tilde{p}(\mathbf{o}_n, \mathbf{a}_n|z)}\log p_{\boldsymbol{\theta}_z}(\mathbf{a}_n|\mathbf{o}_n, z).$$

Noting that $\sum_n \tilde{p}(\mathbf{o}_n, \mathbf{a}_n|z)$ is a constant scaling factor concludes the derivation.

## C  Experiment Setup

### C.1  Environments and Datasets

#### C.1.1  Obstacle Avoidance

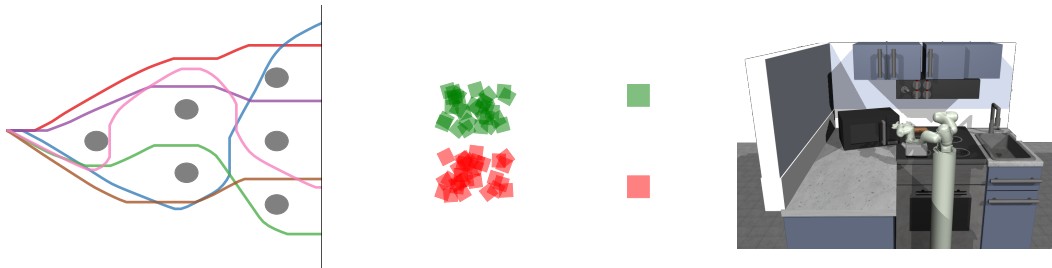

Figure 7: The left figure shows 6 out of 24 ways of completing the obstacle avoidance task. The middle figure shows 30 randomly sampled initial block configurations for the block pushing task. The right figure visualizes the Franka kitchen environment.

**Dataset.** The obstacle avoidance dataset contains 96 trajectories resulting in a total of 7.3k $(\mathbf{o}, \mathbf{a})$ pairs. The observations $\mathbf{o} \in \mathbb{R}^4$ contain the end-effector position and velocity in Cartesian space. Please note that the height of the robot is fixed. The actions $\mathbf{a} \in \mathbb{R}^2$ represent the desired position of the robot. The data is recorded such that there are an equal amount of trajectories for all 24 ways of avoiding the obstacles and reaching the target line. For successful example trajectories see Figure 7.

**Performance Metrics.** The *success rate* indicates the number of end-effector trajectories that successfully reach the target line (indicated by green color in Figure 2). The *entropy*

$$\mathcal{H}(\beta) = -\sum_{\beta} p(\beta) \log_{24} p(\beta),$$

is computed for successful trajectories, where each behavior $\beta$ is one of the 24 ways of completing the task. To assess the model performance, we simulate 1000 end-effector trajectories. We count the number of successful trajectories for each way of completing the task. From that, we calculate a categorical distribution over behaviors $p(\beta)$ which is used to compute the entropy. By the use of $\log_{24}$ we make sure that $\mathcal{H}_{24}(\boldsymbol{\tau}) \in [0, 1]$. If a model is able to discover all modes in the data distribution with equal probability, its entropy will be close to 1. In contrast, $\mathcal{H}(\beta) = 0$ if a model only learns one solution.

#### C.1.2  Block Pushing

**Dataset.** The block pushing dataset contains 500 trajectories for each of the four push sequences (see Figure 8) resulting in a total of 2000 trajectories or 463k $(\mathbf{o}, \mathbf{a})$ pairs. The observations $\mathbf{o} \in \mathbb{R}^{16}$ contain the desired position and velocity of the robot in addition to the position and orientation of the green and red block. Please note that the orientation of the blocks is represented as quaternion number system and that the height of the robot is fixed. The actions $\mathbf{a} \in \mathbb{R}^2$ represent the desired position of the robot. This task is similar to the one proposed in [17]. However, they use a deterministic controller to record the data whereas we use human demonstrators which increases the difficulty of the task significantly due to the inherent versatility in human behavior.

**Performance Metrics.** The *success rate* indicates the number of end-effector trajectories that successfully push both blocks to different target zones. To assess the model performance on non-successful trajectories, we consider the *distance error*, that is, the Euclidean distance from the blocks to the target zones at the final block configuration of an end-effector trajectory. As there are a total of four push sequences $\beta$ (see Figure 2) we use the expected *entropy*

$$\mathbb{E}_{p(\mathbf{o}_0)}\big[\mathcal{H}(\beta|\mathbf{o}_0)\big] \approx -\frac{1}{N_0} \sum_{\mathbf{o}_0 \sim p(\mathbf{o}_0)} \sum_{\beta} p(\beta|\mathbf{o}_0) \log_4 p(\beta|\mathbf{o}_0),$$

to quantify a model's ability to cover the modes in the data distribution. Please note that we use $\log_4$ for the purpose of enhancing interpretability, as it ensures $\mathcal{H}(\beta) \in [0, 1]$. An entropy value

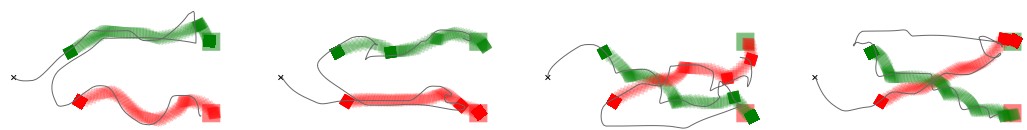

Figure 8: **Block pushing:** Top view of four different push sequences. Starting from the black cross, the gray line visualizes the end-effector trajectory of the robot manipulator. The small rectangles indicate different box configurations in the push sequence, the big rectangles mark the target zones.

of 0 signifies a policy that consistently executes the same behavior, while an entropy value of 1 represents a diverse policy that executes all behaviors with equal probability and hence matches the true behavior distribution by design of the data collection process. Furthermore, we set $p(\mathbf{o}_0) = 1/30$ as we sample 30 block configurations uniformly from a configuration space (see Figure 7). For each $\mathbf{o}_0$ we simulate 16 end-effector trajectories. For a given configuration, we count how often each of the four push-sequences is executed successfully and use the result to calculate a categorical distribution $p(\beta|\mathbf{o}_0)$. Once repeated for all 30 configurations , we compute $\mathbb{E}_{p(\mathbf{o}_0)}\big[\mathcal{H}(\beta|\mathbf{o}_0)\big]$.

### C.1.3   Table Tennis

**Dataset.** The table tennis dataset contains $5000$ $(\mathbf{o}, \mathbf{a})$ pairs. The observations $\mathbf{o} \in \mathbb{R}^4$ contain the coordinates of the initial and target ball position as projection on the table. Movement primitives (MPs) [31] are used to describe the joint space trajectories of the robot manipulator using two basis functions per joint and thus $\mathbf{a} \in \mathbb{R}^{14}$.

**Metrics.** To evaluate the different algorithms on the demonstrations recorded using the table tennis environment quantitatively, we employ two performance metrics: The *success rate* and the *distance error*. The success rate is the percentage of strikes where the ball is successfully returned to the opponent's side. The distance error, is the distance between the target position and landing position of the ball for successful strikes.

### C.1.4   Human Subjects for Data Collection

For the obstacle avoidance as well as the block pushing experiments we used data collected by humans. We note that all human subjects included in the data collection process are individuals who are collaborating on this work. The participants did, therefore, not receive any financial compensation for their involvement in the study.

### C.2   IMC Details and Hyperparameter

IMC employs a parameterized inference network and conditional Gaussian distributions to represent experts. For the latter, we use a fixed variance of 1 and parameterize the means as neural networks. For both inference network and expert means we use residual MLPs [57]. For all experiments, we use batch-size $|\mathcal{B}| = |\mathcal{D}|$, number of components $N_z = 50$ and expert learning rate equal to $5 \times 10^{-4}$. Furthermore, we initialized all curriculum weights as $p(\mathbf{o}_n, \mathbf{a}_n|z) = 1$. For the table tennis and obstacle avoidance task, we found the best results using a multi-head expert parameterization (see Section E.2) where we tested $1 - 4$ layer neural networks. We found that using 1 layer with 32 neurons performs best on the table tennis task and 2 layer with 64 neurons for the obstacle avoidance task. For the block pushing and Franka kitchen experiments, we obtained the best results using a sigle-head parameterization of the experts. We used 6 layer MLPs with 128 neurons for both tasks. For the inference network, we used a fixed set of parameters that are listed in Table 2. For the entropy scaling factor $\eta$ we performed a hyperparameter sweep using Bayesian optimization. The respective values are $\eta = 1/30$ for obstacle avoidance, $\eta = 2$ for block pushing and Franka kitchen and $\eta = 1$ for table tennis.

### C.3   Baselines and Hyperparameter

We now briefly mention the baselines and their hyperparameters. We used Bayesian optimization to tune the most important hyperparameters.

Table 2: **IMC & EM Hyperparameter.**

| PARAMETER | VALUE |
|---|---|
| EXPERT LEARNING RATE | $10^{-4}$ |
| EXPERT BATCHSIZE | 1024 |
| EXPERT VARIANCE ($\sigma^2$) | 1 |
| INFERENCE NET HIDDEN LAYER | 6 |
| INFERENCE NET HIDDEN UNITS | 256 |
| INFERENCE NET EPOCHS | 800 |
| INFERENCE NET LEARNING RATE | $10^{-3}$ |
| INFERENCE NET BATCHSIZE | 1024 |

**Mixture of Experts trained with Expectation-Maximization (EM).** The architecture of the mixture of experts model trained with EM [48] is identical to the one optimized with IMC: We employ a parameterized inference network and conditional Gaussian distributions to represent experts with the same hyperparameters as shown in Table 2. Furthermore, we initialized all responsibilities as $p(z|\mathbf{o}) = 1/N_z$, where $N_z$ is the number of components.

**Mixture Density Network (MDN).** The mixture density network [8] uses a shared backbone neural network with multiple heads for predicting component indices as well as the expert likelihood. For the experts, we employ conditional Gaussians with a fixed variance. The model likelihood is maximized in an end-to-end fashion using stochastic gradient ascent. We experimented with different backbones and expert architectures. However, we found that the MDN is not able to partition the input space in a meaningful way, often resulting in sub-optimal outcomes, presumably due to mode averaging. To find an appropriate model complexity we tested up to 50 expert heads. We found that the number of experts heads did not significantly influence the results, further indicating that the MDN is not able to utilize multiple experts to solve sub-tasks. We additionally experimented with a version of the MDN that adds an entropy bonus to the objective [58] to encourage more diverse and multimodal solutions. However, we did not find significant improvements compared to the standard version of the MDN. For a list of hyperparameter choices see 3.

Table 3: **MDN Hyperparameter.** The 'Value' column indicates sweep values for the obstacle avoidance task, the block pushing task, the Franka kitchen task and the table tennis task (in this order).

| PARAMETER | SWEEP | VALUE |
|---|---|---|
| EXPERT HIDDEN LAYER | $\{1, 2\}$ | $1, 1, 1, 1$ |
| EXPERT HIDDEN UNITS | $\{30, 50\}$ | $50, 30, 30, 50$ |
| BACKBONE HID. LAYER | $\{2, 3, 4, 6, 8, 10\}$ | $3, 2, 4, 3$ |
| BACKBONE HID. UNITS | $\{50, 100, 150, 200\}$ | $200, 200, 200, 200$ |
| LEARNING RATE $\times 10^{-3}$ | $[0.1, 1]$ | $5.949, 7.748, 1.299, 2.577$ |
| EXPERT VARIANCE ($\sigma^2$) | $-$ | 1 |
| MAX. EPOCHS | $-$ | 2000 |
| BATCHSIZE | $-$ | 512 |

**Denoising Diffusion Probabilistic Models (DDPM).** We consider the denoising diffusion probabilistic model proposed by [24]. Following common practice we parameterize the model as residual MLP [27] with a sinusoidal positional encoding [54] for the diffusion steps. Moreover, we use the cosine-based variance scheduler proposed by [59]. For further details on hyperparameter choices see Table 4.

**Normalizing Flow (NF).** For all experiments, we build the normalizing flow by stacking masked autoregressive flows [60] paired with permutation layers [18]. As base distribution, we use a conditional isotropic Gaussian. Following common practice, we optimize the model parameters by maximizing its likelihood. See Table 5 for a list of hyperparameters.

**Conditional Variational Autoencoder (CVAE).** We consider the conditional version of the autoencoder proposed in [20]. We parameterize the encoder and decoder with a neural network with mirrored

Table 4: **DDPM Hyperparameter.** The 'Value' column indicates sweep values for the obstacle avoidance task, the block pushing task, the Franka kitchen task, and the table tennis task (in this order).

| PARAMETER | SWEEP | VALUE |
|---|---|---|
| HIDDEN LAYER | $\{4, 6, 8, 10, 12\}$ | $6, 6, 8, 6$ |
| HIDDEN UNITS | $\{50, 100, 150, 200\}$ | $200, 150, 200, 200$ |
| DIFFUSION STEPS | $\{5, 15, 25, 50\}$ | $15, 15, 15, 15$ |
| VARIANCE SCHEDULER | — | COSINE |
| LEARNING RATE | — | $10^{-3}$ |
| MAX. EPOCHS | — | 2000 |
| BATCHSIZE | — | 512 |

Table 5: **NF Hyperparameter.** The 'Value' column indicates sweep values for the obstacle avoidance task, the block pushing task, the Franka kitchen task and the table tennis task (in this order).

| PARAMETER | SWEEP | VALUE |
|---|---|---|
| NUM. FLOWS | $\{4, 6, 8, 10, 12\}$ | $6, 6, 4, 4$ |
| HIDDEN UNITS PER FLOW | $\{50, 100, 150, 200\}$ | $100, 150, 200, 150$ |
| LEARNING RATE $\times 10^{-4}$ | $[0.01, 10]$ | $7.43, 4.5, 4.62, 7.67$ |
| MAX. EPOCHS | — | 2000 |
| BATCHSIZE | — | 512 |

architecture. Moreover, we consider an additional scaling factor ($\beta$) for the KL regularization in the lower bound objective of the VAE as suggested in [61].

Table 6: **CVAE Hyperparameter.** The 'Value' column indicates sweep values for the obstacle avoidance task, the block pushing task, the Franka kitchen task and the table tennis task (in this order).

| PARAMETER | SWEEP | VALUE |
|---|---|---|
| HIDDEN LAYER | $\{4, 6, 8, 10, 12\}$ | $8, 10, 4, 4$ |
| HIDDEN UNITS | $\{50, 100, 150, 200\}$ | $100, 150, 100, 100$ |
| LATENT DIMENSION | $\{4, 16, 32, 64\}$ | $32, 16, 16, 16$ |
| $D_{\text{KL}}$ SCALING ($\beta$) | $[10^{-3}, 10^{2}]$ | $1.641, 1.008, 0.452, 0.698$ |
| LEARNING RATE | — | $10^{-3}$ |
| MAX. EPOCHS | — | 2000 |
| BATCHSIZE | — | 512 |

**Implicit Behavior Cloning (IBC).** IBC was proposed in [17] and uses energy-based models to learn a joint distribution over inputs and targets. Following common practice we parameterize the model as neural network. Moreover, we use the version that adds a gradient penalty to the InfoNCE loss [17]. For sampling, we use gradient-based Langevin MCMC [57]. Despite our effort, we could not achieve good results with IBC. A list of hyperparameters is shown in Table 7.

**Behavior Transformer (BET).** Behavior transformers were recently proposed in [22]. The model employs a minGPT transformer [62] to predict targets by decomposing them into cluster centers and residual offsets. To obtain a fair comparison, we compare our method to the version with no history. A comprehensive list of hyperparameters is shown in Table 8.

Table 7: **IBC Hyperparameter.** The 'Value' column indicates sweep values for the obstacle avoidance task and the table tennis task (in this order). We do not get any good results for the block push task and the Franka kitchen task.

| PARAMETER | SWEEP | VALUE |
|---|---|---|
| HIDDEN DIM | $\{50, 100, 150, 200, 256\}$ | $200, 256$ |
| HIDDEN LAYERS | $\{4, 6, 8, 10\}$ | $4, 6$ |
| NOISE SCALE | $[0.1, 0.5]$ | $0.1662, 0.1$ |
| TRAIN SAMPLES | $[8, 64]$ | $44, 8$ |
| NOISE SHRINK | — | $0.5$ |
| TRAIN ITERATIONS | — | $20$ |
| INFERENCE ITERATIONS | — | $40$ |
| LEARNING RATE | — | $10^{-4}$ |
| BATCH SIZE | — | $512$ |
| EPOCHS | — | $1000$ |

Table 8: **BET Hyperparameter.** The 'Value' column indicates sweep values for the obstacle avoidance task, the block pushing task, the Franka kitchen task and the table tennis task (in this order).

| PARAMETER | SWEEP | VALUE |
|---|---|---|
| TRANSFORMER BLOCKS | $\{2, 3, 4, 6\}$ | $3, 4, 6, 2$ |
| OFFSET LOSS SCALE | $\{1.0, 100.0, 1000.0\}$ | $1.0, 1.0, 1.0, 1.0$ |
| EMBEDDING WIDTH | $\{48, 72, 96, 120\}$ | $96, 72, 120, 48$ |
| NUMBER OF BINS | $\{8, 10, 16, 32, 50, 64\}$ | $50, 10, 64, 64$ |
| ATTENTION HEADS | $\{4, 6\}$ | $4, 4, 6, 4$ |
| CONTEXT SIZE | — | $1$ |
| TRAINING EPOCHS | — | $500$ |
| BATCH SIZE | — | $512$ |
| LEARNING RATE | — | $10^{-4}$ |

# D  Extended Related Work

## D.1  Connection to ML-Cur

This section elaborates on the differences between this work and the work by [30], as both studies leverage curriculum learning paired with Mixture of Experts to facilitate the acquisition of diverse skills. However, it's essential to discern several noteworthy distinctions between our approach and theirs:

Firstly, Li et al. primarily focus on linear policies and a gating distribution with limited flexibility, constraining the expressiveness of their learned policies. In contrast, our approach allows for the use of arbitrarily non-linear neural network parameterizations for both, expert policies and gating.

Another divergence lies in the handling of mini-batches during training. While our algorithm accommodates the use of mini-batches, Li et al.'s method does not support this feature. The ability to work with mini-batches can significantly enhance the efficiency and scalability of the learning process, especially when dealing with extensive datasets or intricate environments.

Additionally, Li et al.'s evaluation primarily centers around relatively simple tasks, that do not require complex manipulations, indicating potential limitations in terms of task complexity and applicability. In contrast, our work is designed to address more intricate and challenging environments, expanding the range of potential applications and domains.

Lastly, it's worth noting that Li et al. specifically focus on the learning of skills parameterized by motion primitives [31]. In contrast, our framework offers the versatility to encompass various skill types and parameterizations, broadening the scope of potential applications and use cases.

## D.2 Connection to Expectation Maximization

In this section we want to highlight the commonalities and differences between our algorithm and the expectation-maximization (EM) algorithm for mixtures of experts. First, we look at the updates of the variational distribution $q$. Next, we compare the expert optimization. Lastly, we take a closer look at the optimization of the gating distribution.

The EM algorithm sets the variational distribution during the E-step to

$$q(z|\mathbf{o}_n) = p(z|\mathbf{o}_n, \mathbf{a}_n) = \frac{p_{\boldsymbol{\theta}_z}(\mathbf{a}_n|\mathbf{o}_n, z)p(z|\mathbf{o}_n)}{\sum_z p_{\boldsymbol{\theta}_z}(\mathbf{a}_n|\mathbf{o}_n, z)p(z|\mathbf{o}_n)}, \tag{7}$$

for all samples $n$ and components $z$. In the M-step, the gating distribution $p(z|\mathbf{o})$ is updated such that the KL divergence between $q(z|\mathbf{o})$ and $p(z|\mathbf{o})$ is minimized. Using the properties of the KL divergence, we obtain a global optimum by setting $p(z|\mathbf{o}_n) = q(z|\mathbf{o}_n)$ for all $n$ and all $z$. This allows us to rewrite Equation 7 using the recursion in $q$, giving

$$q(z|\mathbf{o}_n)^{(i+1)} = \frac{p_{\boldsymbol{\theta}_z}(\mathbf{a}_n|\mathbf{o}_n, z)q(z|\mathbf{o}_n)^{(i)}}{\sum_z p_{\boldsymbol{\theta}_z}(\mathbf{a}_n|\mathbf{o}_n, z)q(z|\mathbf{o}_n)^{(i)}},$$

where $(i)$ denotes the iteration of the EM algorithm. The update for the variational distribution of the IMC algorithm is given by

$$q(z|\mathbf{o}_n, \mathbf{a}_n)^{(i+1)} = \frac{\tilde{p}(\mathbf{o}_n, \mathbf{a}_n|z)^{(i+1)}}{\sum_z \tilde{p}(\mathbf{o}_n, \mathbf{a}_n|z)^{(i+1)}} = \frac{p_{\boldsymbol{\theta}_z}(\mathbf{a}_n|\mathbf{o}_n, z)^{1/\eta}q(z|\mathbf{o}_n, \mathbf{a}_n)^{(i)}}{\sum_z p_{\boldsymbol{\theta}_z}(\mathbf{a}_n|\mathbf{o}_n, z)^{1/\eta}q(z|\mathbf{o}_n, \mathbf{a}_n)^{(i)}}.$$

Consequently, we see that $q(z|\mathbf{o}) = q(z|\mathbf{oa})$ for $\eta = 1$. However, the two algorithms mainly differ in the M-step for the experts: The EM algorithm uses the variational distribution to assign weights to samples, i.e.

$$\max_{\boldsymbol{\theta}_z} \sum_{n=1}^{N} q(z|\mathbf{o}_n) \log p_{\boldsymbol{\theta}_z}(\mathbf{a}_n|\mathbf{o}_n, z),$$

whereas IMC uses the curricula as weights, that is,

$$\max_{\boldsymbol{\theta}_z} \sum_{n=1}^{N} p(\mathbf{o}_n, \mathbf{a}_n|z) \log p_{\boldsymbol{\theta}_z}(\mathbf{a}_n|\mathbf{o}_n, z).$$

This subtle difference shows the properties of moment and information projection: In the EM algorithm each sample $\mathbf{o}_n$ contributes to the expert optimization as $\sum_z q(z|\mathbf{o}_n) = 1$. However, if all curricula ignore the $n$th sample, it will not have impact on the expert optimization. Assuming that the curricula ignore samples that the corresponding experts are not able to represent, IMC prevents experts from having to average over 'too hard' samples. Furthermore, this results in reduced outlier sensitivity as they are likely to be ignored for expert optimization. Lastly, we highlight the difference between the gating optimization: Assuming that both algorithms train a gating network $g_\phi(z|\mathbf{o})$ we have

$$\max_{\boldsymbol{\phi}} \sum_n \sum_z q(z|\mathbf{o}_n) \log g_\phi(z|\mathbf{o}_n),$$

for the EM algorithm and

$$\max_{\boldsymbol{\phi}} \sum_n \sum_z \tilde{p}(\mathbf{o}_n, \mathbf{a}_n|z) \log g_\phi(z|\mathbf{o}_n),$$

for IMC. Similar to the expert optimization, EM includes all samples to fit the parameters of the gating network, whereas IMC ignores samples where the unnormalized curriculum weights $\tilde{p}(\mathbf{o}_n, \mathbf{a}_n|z)$ are zero for all components.

# E  Algorithm Details & Ablation Studies

## E.1 Inference Details

We provide pseudocode to further clarify the inference procedure of our proposed method (see Algorithm 2).

**Algorithm 2** IMC Action Generation
___
1: **Require:** Curriculum weights $\{\tilde{p}(\mathbf{o}_n, \mathbf{a}_n | z) \mid n \in \{1, ..., N\}, z \in \{1, ..., N_z\}\}$
2: **Require:** Expert parameter $\{\boldsymbol{\theta}_z \mid z \in \{1, ..., N_z\}\}$
3: **Require:** New observation $\mathbf{o}^*$
4: **if not** parameter_updated **then**
5: $\quad \boldsymbol{\phi}^* \leftarrow \arg\max_{\boldsymbol{\phi}} \sum_n \sum_z \tilde{p}(\mathbf{o}_n, \mathbf{a}_n | z) \log g_{\boldsymbol{\phi}}(z | \mathbf{o}_n)$
6: $\quad$ parameter_updated $\leftarrow$ True
7: **end if**
8: Sample $z' \sim g_{\boldsymbol{\phi}^*}(z | \mathbf{o}^*)$
9: Sample $\mathbf{a}' \sim p_{\boldsymbol{\theta}_z}(\mathbf{a} | \mathbf{o}^*, z')$
10: **Return** $\mathbf{a}'$
___

## E.2 Expert Design Choices

**Distribution.** In our mixture of experts policy, we employ Gaussian distributions with a fixed variance to represent the individual experts. This choice offers several benefits in terms of likelihood calculation, optimization and ease of sampling:

To perform the M-Step for the curricula (Section 3.3), exact log-likelihood computation is necessary. This computation becomes straightforward when using Gaussian distributions. Additionally, when Gaussian distributions with fixed variances are employed to represent the experts, the M-Step for the experts simplifies to a weighted squared-error minimization. Specifically, maximizing the weighted likelihood reduces to minimizing the weighted squared error between the predicted actions and the actual actions.

The optimization problem for the expert update can be formulated as follows:

$$\boldsymbol{\theta}_z^* = \arg\max_{\boldsymbol{\theta}_z} \sum_n \tilde{p}(\mathbf{o}_n, \mathbf{a}_n | z) \log p_{\boldsymbol{\theta}_z}(\mathbf{a}_n | \mathbf{o}_n, z) = \arg\min_{\boldsymbol{\theta}_z} \sum_n \tilde{p}(\mathbf{o}_n, \mathbf{a}_n | z) \| \boldsymbol{\mu}_{\boldsymbol{\theta}_z}(\mathbf{o}_n) - \mathbf{a}_n \|_2^2.$$

This optimization problem can be efficiently solved using gradient-based methods. Lastly, sampling from Gaussian distributions is well-known to be straightforward and efficient.

**Parameterization.** We experimented with two different parameterizations of the Gaussian expert means $\boldsymbol{\mu}_{\boldsymbol{\theta}_z}$, which we dub *single-head* and *multi-head*: For single-head, there is no parameter sharing between the different experts. Each expert has its own set of parameters $\boldsymbol{\theta}_z$. As a result, we learn $N_z$ different multi-layer perceptrons (MLPs) $\boldsymbol{\mu}_{\boldsymbol{\theta}_z} : \mathbb{R}^{|\mathcal{O}|} \to \mathbb{R}^{|\mathcal{A}|}$, where $N_z$ is the number of mixture components. In contrast, the multi-head parameterization uses a global set of parameters $\boldsymbol{\theta}$ for all experts and hence allows for feature sharing. We thus learn a single MLP $\boldsymbol{\mu}_{\boldsymbol{\theta}} : \mathbb{R}^{|\mathcal{O}|} \to \mathbb{R}^{N_z \times |\mathcal{A}|}$.

To compare both parameterizations, we conducted an ablation study where we evaluate the MoE policy on obstacle avoidance, table tennis and Franka kitchen. In order to have a similar number of parameters, we used smaller MLPs for single-head, that is, $1 - 4$ layers whereas for multi-head we used a 6 layer MLP. The results are shown in Table 9 and are generated using 30 components for the obstacle avoidance and table tennis task. The remaining hyperparameters are equal to the ones listed in the main manuscript. For Franka kitchen, we report the cumulative success rate and entropy for a different number of completed tasks. We report the mean and standard deviation calculated across 10 different seeds. Our findings indicate that, in the majority of experiments, the single-head parameterization outperforms the mutli-head alternative. Notably, we observed a substantial performance disparity, especially in the case of Franka kitchen.

Table 9: **Expert Parameterization Ablation**: We compare IMC with single- and multi-head expert parameterization. For further details, please refer to the accompanying text.

| ARCHITECTURE | OBSTACLE AVOIDANCE | | TABLE TENNIS | | FRANKA KITCHEN | |
| --- | --- | --- | --- | --- | --- | --- |
| | SUCCESS RATE (↑) | ENTROPY (↑) | SUCCESS RATE (↑) | DISTANCE ERR. (↓) | SUCCESS RATE (↑) | ENTROPY (↑) |
| SINGLE-HEAD | $0.899_{\pm 0.035}$ | $0.887_{\pm 0.043}$ | $0.812_{\pm 0.039}$ | $0.168_{\pm 0.007}$ | $3.644_{\pm 0.230}$ | $6.189_{\pm 1.135}$ |
| MULTI-HEAD | $0.855_{\pm 0.053}$ | $0.930_{\pm 0.031}$ | $0.870_{\pm 0.017}$ | $0.153_{\pm 0.007}$ | $3.248_{\pm 0.062}$ | $4.657_{\pm 0.312}$ |

**Expert Complexity.** We conducted an ablation study to evaluate the effect of expert complexity on the performance of the IMC algorithm. The study involved varying the number of hidden layers in the single-head expert architecture while assessing the IMC algorithm's performance on the Franka kitchen task using the cumulative success rate and entropy. The results, presented in Figure 9, were obtained using the hyperparameters specified in the main manuscript. Mean and standard deviation were calculated across 5 different seeds. Our findings demonstrate a positive correlation between expert complexity and achieved performance.

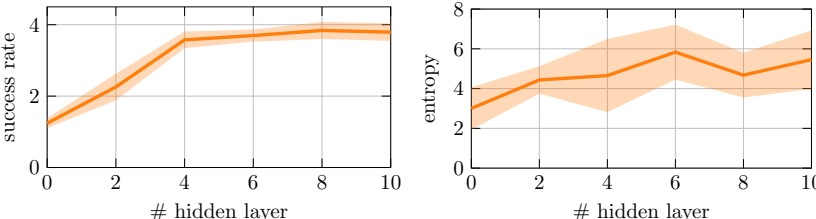

Figure 9: **Expert Complexity Ablation:** Evaluation of the IMC algorithm on the Franka kitchen task with varying numbers of hidden layers in the single-head expert architecture.

### E.3 Component Utilization

We further analyze how IMC utilizes its individual components. Specifically, we assess the entropy of the weights assigned to these components, denoted as $\mathcal{H}(z)$ and defined as $\mathcal{H}(z) = -\sum_z p(z) \log p(z)$. Maximum entropy occurs when the model allocates equal importance to all components to solve a task, which implies that $p(z) = 1/N_z$. Conversely, if the model relies solely on a single component, the entropy $\mathcal{H}(z)$ equals zero. We conduct the study on the Franka kitchen task, evaluating it through cumulative success rates and entropy. The results are shown in Figure 10a. We computed the mean and standard deviation based on data collected from 5 different seeds. Our findings reveal that IMC exhibits substantial component utilization even when dealing with a large number of components, denoted as $N_z$.

### E.4 Number of Components Sensitivity

To investigate how the algorithm responds to changes in the number of components $N_z$, we performed a comprehensive ablation study. Figure 10b shows the success rate of IMC on the Franka kitchen task using a varying number of components. Our findings indicate that the algorithm's performance remains stable across different values of $N_z$. This robust behavior signifies that the algorithm can adeptly accommodate varying numbers of components without experiencing substantial performance fluctuations. Hence, opting for a higher number of components primarily entails increased computational costs, without leading to overfitting issues.

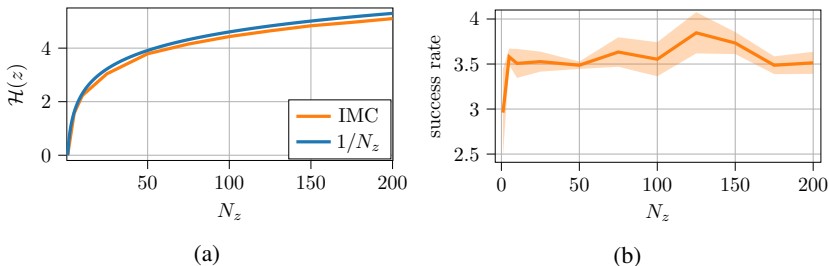

(a)  (b)

Figure 10: Figure 10a shows the component utilization of IMC: The entropy of IMC's mixture weights $p(z)$ is close to the entropy of a uniform distribution $1/N_z$ indicating that the algorithm uses all components for solving a task, even for a high number of components $N_z$.

### E.5 Curriculum Pacing Sensitivity

To examine the algorithm's sensitivity to the curriculum pacing parameter $\eta$, we conducted an ablation study. Figure 11 presents the results obtained using 30 components for the obstacle avoidance and table tennis tasks, while maintaining the remaining hyperparameters as listed in the main manuscript. For the Franka kitchen task, we analyzed the cumulative success rate and entropy across varying numbers of completed tasks. The mean and standard deviation were calculated across 5 different seeds. Our findings reveal that the optimal value for $\eta$ is dependent on the specific task. Nevertheless, the algorithm exhibits stable performance even when $\eta$ values differ by an order of magnitude.

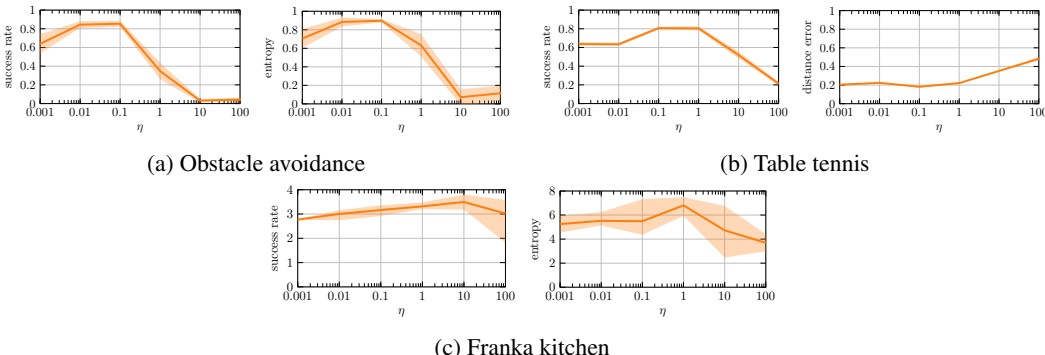

(a) Obstacle avoidance  (b) Table tennis

(c) Franka kitchen

Figure 11: **Curriculum Pacing Sensitivity:** Sensitivity analysis of the IMC algorithm for performance metrics in the obstacle avoidance, table tennis, and Franka kitchen tasks, considering varying curriculum pacing ($\eta$) values. The results illustrate the mean and standard deviation across 5 different seeds.

