# OpenReview forum: "Information Maximizing Curriculum: A Curriculum-Based Approach for Learning Versatile Skills"
_NeurIPS.cc/2023/Conference — NeurIPS 2023 poster_

### Official Review · Reviewer_kzo3 · 2023-07-05

**Soundness:** 3 good
**Presentation:** 3 good
**Contribution:** 3 good
**Rating:** 6
**Confidence:** 3

**Summary:**

This paper proposes an imitation learning method IMC that can model multi-modal behaviors. IMC avoids the mode-averaging issue with an objective similar to reverse-KL. To cover all modes in the dataset, IMC further introduces a mixture model with multiple components, each focusing on different data distribution it specializes to. The mixture model is optimized with the EM algorithm. The authors provide extensive experiments over simulated control environments and demonstrate IMC's superior modeling abilities.


**Strengths:**

1. IMC is a well-motivated method for multi-modal density estimation and provides an elegant reverse KL-based solution.
2. The experiments in low-dimensional control environments are extensive. IMC is compared against major generative models (with maximum likelihood objective) and addresses the mode-averaging issue better.
3. The visualization of learned trajectories clearly demonstrates the learned modes of different methods and the natural mode-ignorance ability of IMC.


**Weaknesses:**

The presentation in the experiment section could be improved. Currently, the subsections for different environments repeat mostly the same conclusions that IMC achieves high success rates and covers diverse modes. I think it is better to emphasize anything special for different environments or different baselines.


**Questions:**

1. How sensitive is the performance of IMC with respect to different choices of $\eta$?
2. Can IMC be scaled up to scenarios with high-dimensional input?


**Limitations:**

The authors have discussed the limitations.

---

> ### Author Rebuttal · Authors · 2023-08-09
>
> > 1. IMC is a well-motivated method for multi-modal density estimation and provides an elegant reverse KL-based solution.
> >  2. The experiments in low-dimensional control environments are extensive. IMC is compared against major generative models (with maximum likelihood objective) and addresses the mode-averaging issue better.
> > 3. The visualization of learned trajectories clearly demonstrates the learned modes of different methods and the natural mode-ignorance ability of IMC.
>
>
> We thank the reviewers for their positive feedback.  We are glad the reviewers recognize our contribution in addressing the mode-averaging problem. We now aim to thoroughly address the concerns raised by the reviewers.
>
> > The presentation in the experiment section could be improved. Currently, the subsections for different environments repeat mostly the same conclusions that IMC achieves high success rates and covers diverse modes. I think it is better to emphasize anything special for different environments or different baselines.
>
> We are grateful to the reviewers for their valuable suggestion. We agree that the content in the experiment section currently provides limited additional insights beyond what's presented in the tables and figures within the paper. Consequently, we are committed to enhancing the experiment section by placing a stronger emphasis on distinct characteristics of various environments.
>
> > How sensitive is the performance of IMC with respect to different choices of $\eta$?
>
> To address your question thoroughly, we have included a comprehensive analysis of IMC's performance sensitivity in the supplementary material of the paper (Appendix E.2). In that section, we discuss the impact of various values of $\eta$ on the model's performance on the Obstacle Avoidance, Table Tennis and Franka Kitchen task.
>
> > Can IMC be scaled up to scenarios with high-dimensional input?
>
> Indeed! IMC can be effectively scaled up to scenarios with high-dimensional input. A viable approach to achieve this scalability is by utilizing an encoder, such as a Convolutional Neural Network (CNN) encoder, to map the high-dimensional inputs (such as images) to a lower-dimensional space before passing them to the inference network or experts.

---

> > ### Comment · Reviewer_kzo3 · 2023-08-17
> >
> > Thank you for your response. I appreciate the newly added sensitivity analysis and discussions. I am keeping my original score.

---

### Official Review · Reviewer_tAUb · 2023-07-05

**Soundness:** 2 fair
**Presentation:** 3 good
**Contribution:** 2 fair
**Rating:** 6
**Confidence:** 3

**Summary:**

The authors study a method, Information Maximizing Curriculum (IMC), that performs behavioral cloning by having the model selectively choose a learned weighing of the demonstration data for which the model is best at predicting (via minimizing the reverse KL divergence). To avoid the mode-seeking behavior of this approach, the authors extend the method to make use of K such model components, leading to a mixture of experts (MoE) approach, whereby each component selectively models the distribution in this way, while maximizing their joint information projection over the dataset (via simultaneously maximizing the entropy of the MoE distribution). Experiments in two robotics  simulators show it outperforms several other baselines based on generative models and basic MoE methods trained via expectation-maximization and backpropagation.

**Strengths:**

- The paper is overall well written. The method and its motivations are clearly communicated.
- The authors compare to several important baselines, spanning generative models, energy-based models, and MoE methods, and show strong performance improvements.

**Weaknesses:**

The main weakness I see is that this work seems almost identical to Li, et al (2023), which proposes largely the same method. Moreover, this prior work looks at a very similar set of environments as this paper. While the authors cite this prior work, they do not directly compare to it. Given the extremely close similarity between the two methods, it seems important to compare to this work to justify the contribution in this paper, which in a sense, is an extension of the method in Li, et al (2023) by incorporating neural networks.

I see two main ways to show improvement over this prior work: The authors can either (1) show their method outperforms the approach in Li, et al (2023) on the tasks studied, or (2) Demonstrate clearly how IMC can scale to environments in which the method of Li, et al (2023) cannot, thereby clearly justifying the strengths of this extension. I imagine neither of these aims is too difficult, but such a comparison seems sorely missing, given the strong similarity between the two works.

Lastly, I find it odd that the authors couch their method as “curriculum learning,” ignore the field of active learning altogether, and proceed to imply that their method is novel in not requiring an a priori difficulty metric for each datapoint. The authors should relate their method to active learning, which is a mature field of study, and most active learning methods can be viewed as “curriculum learning” without any a priori notion of task difficulty.

### References

Maximilian Xiling Li, Onur Celik, Philipp Becker, Denis Blessing, Rudolf Lioutikov, and Gerhard Neumann. Curriculum-based imitation of versatile skills. arXiv preprint arXiv:2304.05171,
2023.

**Questions:**

- This study primarily focuses on continuous control robotics simulators. Do you have a sense of whether IMC can be made to work in discrete control settings, e.g. Atari, chip design, web navigation.
- Are there cases where maximizing mode coverage is disadvantageous? Often demonstration datasets contain a mix of approximately optimal and very suboptimal trajectories. Could IMC lead to learning MoE that perform worse in these settings?

**Limitations:**

This work only focuses on fairly simple continuous control tasks. Scaling this to more complex imitation learning settings (e.g. from pixels or controlling a much more complex action space) as well as discrete control settings would add confidence in the utility of this approach. Relatedly, demonstrating success on these more challenging settings would be a great way to highlight why IMC is an important improvement over the Li, et al (2023) work.

---

> ### Author Rebuttal · Authors · 2023-08-09
>
> > The paper is overall well written. ff.
>
> We thank the reviewers for acknowledging our contribution and are committed to addressing your questions and concerns.
>
> > The main weakness I see is that this work seems almost identical to Li, et al (2023), which proposes largely the same method.
>
> We agree with the reviewers that the method proposed by Li, et al (2023) (ML-Cur) and our work seem very similar. However, we want to point to some of the differences:
>
> - ML-Cur assumes Gaussian curricula. The parameters of these distributions must be updated in every iteration of the algorithm and involve expensive matrix inversions. The gating distribution has limited flexibility due to the Gaussianity assumption. In contrast, IMC uses non-parametric curricula $p(\mathcal{D}|z)$ and trains a gating network ($g_{\phi}(z|\mathcal{D})$) once after training which is more efficient and allows for highly non-linear partitioning of the observation space.
> - ML-Cur does not admit mini-batch updates for the model parameters. Therefore, the method does not scale to large datasets. Conversely, IMC accommodates mini-batch updates (refer to Section 3.5), enabling effective scalability.
> - ML-Cur has to update the mixture weights $p(z)$ in every iteration. IMC has implicit updates due to Proposition 3.1.
> - ML-Cur uses linear conditional Gaussian distributions to represent the experts. In contrast, IMC allows for arbitrary experts where exact likelihood computation is possible. In particular, we used non-linear conditional Gaussian distributions in our experiments.
> - ML-Cur is evaluated on *episodic* tasks, i.e., where the model has the learn a mapping from a context to movement primitive parameters which represent the whole trajectory. A much more common setting is where the model has to learn a mapping from observations to actions (*step-based*). In that case, a trajectory is formed by performing multiple steps in the environment of a given task. In our work, 3 out of the 4 tasks consider the latter setup.
>
> > Moreover, this prior work looks at a very similar set of environments as this paper.
> The tasks used in the work of Li, et al (2023)
>
> Despite the similarity in appearance of the environments used in the work of Li, et al and our work, there exist major differences:
>
> - Their tasks do not require complex manipulations of objects in contrast to the Block Pushing task as well as Franka Kitchen used in our work.
> - They consider input spaces with very low dimensions ($<4$) whereas our work considers up to $30$ (Franka Kitchen)
> - They focus on learning from a few data points ($<5000$) while our method looks at datasets with up to $463k$ samples (Block Pushing).
> - Their environments require little generalization capabilities of the models (this can be seen from the highly competitive results achieved by the k-nearest neighbor algorithm). Our environments are much more complex, see e.g. the low performance of behavior transformers (BET) and denoising diffusion models (DDPM) on Obstacle Avoidance, Block Pushing, or Franka Kitchen.
> - They do not provide performance metrics for quantifying how versatile the learned policy is. In contrast, we provide such a metric for all environments except for the table tennis task.
>
> > While the authors cite this prior work, they do not directly compare to it. ff.
>
> We agree with the reviewers that the comparison to the work of Li, et al (ML-Cur) is missing. Therefore, we added quantitative and qualitative comparisons between IMC and ML-Cur in the 'global comment' of the rebuttal and in the accompanying PDF file. We thank the reviewers and will include these results in the final version of the paper.
>
> > I see two main ways to show improvement over this prior work: The authors can either (1) show their method outperforms the approach in Li, et al (2023) on the tasks studied, or (2) Demonstrate clearly how IMC can scale to environments in which the method of Li, et al (2023) cannot ff.
>
> We thank the reviewers for their suggestions. We believe that by including the comparison to ML-Cur (see last comment) in the paper we demonstrate (1) that our method significantly outperforms ML-Cur on the tasks studied and (2) show ML-Cur is not able to scale to the environments used in our work.
>
> > Lastly, I find it odd that the authors couch their method as “curriculum learning,” ignore the field of active learning altogether. ff.
>
> We agree that we should relate our method to active learning. To that end, we will include a section in Chapter 4, elaborating on the commonalities and differences between our work and active learning.
>
> > This study primarily focuses on continuous control robotics simulators. Do you have a sense of whether IMC can be made to work in discrete control settings, e.g. Atari, chip design, web navigation.
>
> Absolutely! While this study primarily focuses on continuous control robotics simulators, our algorithm is applicable for training mixture of experts models whenever the likelihood of the experts can be defined and optimized, including the discrete control setting.
>
> > Are there cases where maximizing mode coverage is disadvantageous? ff.
>
> Yes, we believe there are cases where maximizing mode coverage can be disadvantageous. However, this is a problem of divergence minimization in general and we believe that other methods for learning MoE could lead to even worse performances, as they aim to cover all data even if the model complexity is not sufficient (property of the forward KL). There are dedicated fields of study such as offline reinforcement learning [1], that address problems associated with learning from suboptimal data.
>
> [1] Levine, Sergey, et al. "Offline reinforcement learning: Tutorial, review, and perspectives on open problems.”
>
> We sincerely appreciate the time and effort the reviewers have invested in assessing our work. Their comments and suggestions have significantly contributed to refining and strengthening the manuscript.

---

> ### Comment · Reviewer_tAUb · 2023-08-11
> **Nice additions**
>
> The new experimental comparison to Li et al satisfies my original critiques. I encourage the authors to emphasize the similarity and differences between their work and Li et al in their final manuscript, as the differences articulated above form the *primary contribution* of their work. I am raising my score in response.

---

### Official Review · Reviewer_EzYZ · 2023-07-05

**Soundness:** 3 good
**Presentation:** 3 good
**Contribution:** 2 fair
**Rating:** 6
**Confidence:** 3

**Summary:**

This paper proposes a curriculum based approach for imitation learning. Overall, the imitation learning problem is posed as a conditional density estimation problem. Given the multi-modal nature of underlying data, this paper proposes to learn a curriculum based mixture of expert policy. Intuitively, each expert is only responsible for learning a subset of the training data and learns the policy for this subset using reverse KL. Experimentally, the proposed approach is verified in 4 different environments and compared against both common and recently proposed approaches.

**Strengths:**

The paper presents an interesting and grounded approach for learning from multimodal data distribution. The overall idea of using a curriculum to weight data samples based on how well they are represented under the expert policy seems to be generally useful. The approach is shown to work in diverse settings and seems competitive with recent results.

**Weaknesses:**

*Baseline results:* Overall, the proposed approach provides very little significant advantage over simpler baselines. For instance, in Figure 4 and Figure 5 we can see that for most tasks (Pusing, Tennis, Kitchen), a diffusion model based approach (DDPM) is highly competitive to the proposed approach (success rate difference is less than 0.05) across all tasks. Only in the Obstacle avoidance task, does DDPM approach suffer although CVAE still performs quite well (unclear why DDPM performs poorly here).

*Benchmark tasks:* While the paper tries to evaluate the proposed approach on multiple datasets, most of the tasks/datasets are not commonly used across multimodal tasks (only Franka-Kitchen is the commonly used dataset). Given the large set of recent works in this area it would be much better to evaluate on tasks which other recent works evaluate. For instance, most recent works evaluate on RoboMimic dataset [1] and the block pushing task from IBC [2]. Both of these task suites have human demonstrations available and since the underlying data distribution is highly important for these proposed approaches, reusing datasets from previously proposed approaches will allow for a much fairer comparison.

[1] Robomimic: https://github.com/ARISE-Initiative/robomimic
[2] Florence et al. Implicit Behavior Cloning

--- Post Rebuttal ---

I don't see any issues with the paper the provided code also looks reasonable and reproduces the results so I would update my recommendation to Weak Accept.

**Questions:**

Overall the approach is broadly similar to an EM approach with mixture of policies. As noted, the main difference being that the proposed approach uses a curriculum weighted optimization objective to assign weights to samples instead of directly using the variational distribution. While algorithmically, this difference is not large in terms of empirical results the EM based approaches lags behind the proposed approach quite a lot (especially for the obstacle avoidance task). Do the authors have an intuition why the EM approach suffers so much in comparison?

**Limitations:**

Limitations are discussed

---

> ### Author Rebuttal · Authors · 2023-08-09
>
> > The paper presents an interesting and grounded approach for learning from multimodal data distribution. ff.
>
> We thank the reviewers for their feedback on our paper. We are delighted to hear that the overall idea of using a curriculum to weight data samples is well-received. We would like to address address the mentioned weaknesses / questions:
>
> > Baseline results: Overall, the proposed approach provides very little significant advantage over simpler baselines. For instance [...] a diffusion model based approach (DDPM) is highly competitive to the proposed approach ff.
>
> We appreciate the reviewers' comment and the opportunity to discuss the advantages of our approach in comparison to other baselines. Our method indeed offers distinct benefits over alternative techniques, such as DDPM.
>
> One significant advantage lies in the inference efficiency of IMC. DDPM, due to its sequential inference procedure, suffers from longer inference times, which can be impractical for applications with real-time constraints. This limitation becomes especially pronounced when deploying models to real-world systems. In contrast, IMC's inference process requires just two forward passes: one through the inference network for sampling $z$ and another through the corresponding expert. This efficiency is particularly advantageous compared to DDPM's requirement of $N_{\text{diffusion steps}}$ forward passes. Furthermore, we observered that our method requires less than $1/10$th of the training time that is required to train DDPM. We thank the reviewers and will include an ablation study on inference and training times in the camera-ready version of the paper.
>
> Regarding the term 'simpler baselines', we are uncertain about the reviewers' defintion of 'simple'. We understand that baselines, including DDPM, are far from being simple in terms of their design choices and intricacies. DDPM involves non-trivial decisions concerning parameters like the number of diffusion steps, variance scheduling, time embedding, and employs additional code-level optimizations such as using an exponential moving average for the parameters. The same holds for other baselines such as normalizing flows, energy based models and behavior transformers.
>
> > Only in the Obstacle avoidance task, does DDPM approach suffer although CVAE still performs quite well (unclear why DDPM performs poorly here).
>
> We conjecture that DDPM's subpar performance can be attributed to the limited amount of data available for the obstacle avoidance task. IMC, in contrast, appears to circumvent this challenge due to its composition of 'simpler' Gaussian experts, which appear to generalize better on little data.
>
> Regarding CVAE: While CVAE has a high sucessrate, it suffers from low entropy values, i.e., is not able to cover multple modes in the data distribution. This can also be seen in Figure (3). We've empirically observed that a trade-off often exists between achieving high success rates and maintaining adequate entropy levels across most models (see Table 1). However, IMC stands out as an exception by simultaneously achieving high success rates and commendable entropy values.
>
>
> > Benchmark tasks: [...] most recent works evaluate on RoboMimic dataset [1] and the block pushing task from IBC [2]. Both of these task suites have human demonstrations available ff.
>
> While we agree that the RoboMimic datasets are excellent for evaluating a models' performance in complex manipulation tasks, we opted not to use them since a) the tasks within the RoboMimic datasets (lift, can, square, tool hang and transport) lack multimodality, except for the multimodality evident in human demonstrations and b) they do not provide suitable metrics to evaluate a models' ability to cover different modes present in the dataset.
>
> Regarding the block pushing task from IBC: Unfortunately, this dataset is not recorded by humans but uses a scripted policy (Quote IBC paper, Section 4: [...]We evaluate implicit (EBM) and explicit policies on both variants, trained from a dataset of 2,000 demonstrations using a scripted policy that readjusts its pushing direction if the block slips from the end effector.[...]). We found that this task can be solved by simple methods such as behavior cloning (see IBC paper Table 3, MSE). Therefore, we decided to rebuild the task and use human-recorded data which significantly increases the diffuculty due to the inherent versatility in human demonstrations. Furthermore, we want to highlight that we've acknowledged the resemblance between our task and the one introduced in the IBC paper. We provided an explanation for our decision to redevelop the task, as outlined in Appendix C.1.2.
>
> We carefully designed new tasks and metrics and recorded data to circumvent these shortcommings of existing benchmarks such as Robomimic and intend to publish them, as well as additional tasks, in a separate work. This endeavor aims to establish a benchmark for versatile imitation learning, focusing on quantifying a model's capacity to acquire diverse skills.
>
>
> > Overall the approach is broadly similar to an EM [...]. Do the authors have an intuition why the EM approach suffers so much in comparison?
>
> We believe that the primary reason for the relatively poorer performance of EM, compared to IMC, lies in the inherent requirement of EM to account for all data points during training. This constraint forces EM-trained models to handle outliers or challenging samples that might lead to suboptimal log-likelihood outcomes. In contrast, IMC's curriculum-based approach enables the model to focus on samples where experts perform well (in terms of log-likelihood), allowing them to disregard outliers or poorly performing instances, resulting in the observed performance boost.
>
> We extend our gratitude to the reviewers for dedicating their time and effort to evaluating our work. Their valuable comments and suggestions helped enhancing and fortifying the quality of our manuscript.

---

### Official Review · Reviewer_kVDY · 2023-07-07

**Soundness:** 3 good
**Presentation:** 2 fair
**Contribution:** 3 good
**Rating:** 6
**Confidence:** 4

**Summary:**

The paper proposes a learning protocol that learns multiple policies ("skills"), distribution over skills, and a per-skill priority over experience buffer. This is achieved by maximizing a variational lower bound of a certain averaged regularized KL distance. Namely, the objective is a sum of two terms. The first term is a KL distance between the per-skill priority over previously seen actions and a policy, both conditioned on the skill and previously seen observation, and averaged over them. A second term is an entropy of the joint distribution over experience and skills, which acts as a regularizer, forcing it to be as diverse as possible.

In the experiment section, the skill policies share a common backbone and output a mean of an isotropic Gaussian distribution. The paper tests the method over several tasks, such as Obstacle Avoidance, Block Pushing, Franka Kitchen, or Table Tennis. The approach compares (mostly) favorably against seven baseline methods.

**Strengths:**

The paper proposes a simple objective to train multiple policies at once, such that together they cover multiple modes in the dataset. The results show that the method (mostly) performs better than the considered baselines.

**Weaknesses:**

The empirical part could be improved. Namely:
* Performance metrics should be defined in the main body of the paper:
	* There could be a dedicated section for that purpose. This would improve the exposition and allow more discussion about the method in Sections 5.1-5.4.
	* This could also make reading the results (e.g., Table 1) more accessible. For instance, the definition of entropy varies across environments, as shown in Appendix C, which makes the numbers incomparable and possibly draw wrong conclusions.
	* The discussion on the range of entropy values should be moved from the Appendix to the main body. It provides grounding for the numbers in Table 1.
	* Additional place for this and a more in-depth discussion of the results could be achieved by moving Section 2.1 and Figure 1 to the Appendix (it is a well know property of KL). Similarly, some parts of Section 3 could be moved to the Appendix.
* The paper does not provide numbers on how the algorithm mixes between skills, e.g., the entropy of the distribution over components ($p(z)$).
* It seems that each environment required a different setup of the method. This is a limiting factor for the method. The paper, however, does not guide the potential user of the method in choosing the relevant hyperparameters.  For example, estimating how many modes are in the data can be a non-trivial task, and consequently, setting the number of components ($N_z$) selected for each experiment may be non-trivial. The paper has no information regarding the actual values of $N_z$ chosen for each experiment.
* Descriptions in Sections 5.1-5.4 are mainly technical and deal with the setup. The discussion about the results of IMC is limited to one or two sentences that do not add more information than Table 1 or Figure 6. A reader would expect that most of such a section would provide real insights about the method (e.g., what skills were learned, how they were acquired during training, where all modes were discovered, how a distribution that mixes skills looks, etc.).
* Figure 6: Shouldn't we expect that the performance is an increasing function of the number of components (the more modes covered, the higher the objective)?
* There is no mention of what values $\eta$ were chosen.
* Other issues:
	* Table 1: The description is not self-explanatory. What is the selected number of components? What versions of entropy are used? What is the setup of experiments? Etc.
	* Sections 5.1, 5.2, and 5.4 refer to Table 13, which is not present in the paper. Most likely, it should refer to Table 1.
	* Figure 6 precedes Figure 5.

The quality of the technical part of the paper could be improved and more clearly agitated. In particular,
* Section 2.2:
	* There is no definition of $\mathcal O$, $\mathcal A$, $p$, $z$.
	* Is $z$ a continuous or discrete random variable? From the context of the following sections, it would seem that the latter.
* Section 3.1:
	* It seems more clear to write $p(o, a)$ instead of $p(\mathcal D)$.
	* In equation (2), there should be $\mathcal H(p(\mathcal D))$ (or better $\mathcal H(p(o,a))$, see the item above). Additionally $\mathbb E_{p(\mathcal D)}$ could be more transparent if written as $\mathbb E_{o,a\sim p(\mathcal D)}$.
	* The formulas in lines 80, 85, and 86 should be better justified. There is only a cryptic comment in line 81 about optimization in an alternating fashion, which is not justified, nor a suitable reference is given.
	* The description in lines 87-93 is relatively informal and does not refer to the formulas in lines 80, 85, and 86. Additionally, the text uses colloquial terms such as "representational capacity of the policy", "capacity [..] is exhausted", or alludes to the convergence of curriculum, which was not proved.
	* There is no definition of $\mathcal D_n$ (lines 85, 86, 119, 120, 124, 129, 130, etc.). Why not just write $(o_n, a_n)$?
	* In the proof of Proposition 3.1 (Appendix A.1.), the formulas read $p^*(o)$ where the Proposition refers to $p^*(z)$. A similar comment refers to other proofs in Appendix A.
* Section 3.2
	* There is no description of the objective $J(\psi)$ and its lower bound, showing how the individual terms promote desirable behavior.
	* In equation (3), it seems that the entropy term should be equal to $\mathcal H(p(o, a, z))$; otherwise, equation (4) seems not to be valid.
	* It seems that in equation (4) and in the definition of $R_z$, there should be $q(z|o,a)$ (in place of $q(z|o)$ and $q(z|\mathcal D)$, respectively). Additionally, it would make sense to define $R_z$ as a function of $(o, a)$ (instead of $\mathcal D$).
	* Similar comments apply to Sections 3.3-3.5.

Edit: After the Authors' rebuttal, I have raised the score (4->6).

**Questions:**

Consult the Weaknesses section.

**Limitations:**

The paper includes a brief limitations section. What could also be mentioned is that the method requires setting several important hyperparameters (see the review), some of which require non-trivial knowledge about data. The method also was tested on continuous tasks, so a natural question (limitation or future research) would be to ask about performance on discrete domains, such as combinatorial puzzles like chess or video games like Atari benchmark.

---

> ### Author Rebuttal · Authors · 2023-08-09
>
> In order to adhere to the character limit we will jointly address most of the concerns regarding the empirical part and those of the technical part.
>
> **Regarding concerns on the empirical part:**
>
> We agree with the reviewers that moving the definition of the entropy for the different experiments to the main body of the paper increases readability and makes the results more accessible. To that end, we moved Section 2.1 (explanation of KL properties ), Figure 4, and some of the algorithmic details (Section 3.5) to the Appendix. Additionally, we improved the discussion of the results and highlighted unique aspects of different environments instead of repeating similar conclusions.
> Furthermore, we moved parts of the explanation about the hyperparameters of IMC (which encompasses all design choices incl. $\eta$ and $N_z$ for all experiments) from Appendix C.2 to Section 5. We also made Table 1 self-explanatory by elaborating on the setup and on the metrics used. Moreover, we improved the result discussion by highlighting unique aspects of the different tasks and our method.
>
> > The paper does not provide numbers on how the algorithm mixes between skills ff.
>
> First, we want to clarify potential confusion about the relationship between components $z$ and skills learned by the policy. For the following discussion, we assume that the reviewer refers to a skill as a sequence of actions that lead to a successful outcome. Each $z$ employs a curriculum to specialize on a subset of the data. Depending on the model complexity of the expert, this subset could encompass multiple skills or only part of a skill. Hence, there is no one-to-one correspondence between skills and $z$. Therefore, there is no need for knowing the number of skills or modes a-priori. We used $N_z=50$ for all experiments which empirically worked well. To clarify this further, we added a figure to the supplementary PDF that visualizes the curriculum.
>
> We believe that reporting the entropy $\mathcal{H}(p(z))$ provides valuable insights on how many of the components are used to solve a task. We will add an ablation study showing $\mathcal{H}((p(z))$ for various $N_z$ to the appendix.
>
> >Figure 6: Shouldn't we expect that the performance is an increasing function of the number of components ff.
>
> The performance metrics presented in Figure 6, namely the success rate and entropy, differ from the optimization objective $J(\psi)$. While we do observe empirically that adding more components to the model tends to increase $J(\psi)$, the same does not necessarily hold true for the performance metrics.
>
> The reason behind this observation lies in the fact that as we add more components to the model, the overall model complexity increases. Consequently, the model becomes more susceptible to overfitting on the training data. We conjecture that this causes the fluctuations in the success rate in Figure 6a.
>
> **Regarding concerns on the technical part:**
>
> We appreciate the reviewers' feedback regarding the notation for the curriculum $p(\mathcal{D})$. We understand the potential confusion that might arise from this notation. The intention behind using this notation was to emphasize that the curriculum represents a categorical distribution over data points rather than a continuous joint distribution between actions and observations $p(o,a)$.
>
> Upon careful consideration and in light of the existing MoE literature, which commonly employs $(o,a)$ to denote responsibilities as $q(z|o_n,a_n)$, we find the suggested notation to be more aligned and clear. Therefore, we gladly adopt the notation $(o,a)$ in our work. Moreover, we defined $R_z$ as a function of $(o,a)$ and replaced $\mathcal{D}_n$ with $(o_n,a_n)$.
>
> >- The formulas in lines 80, 85, and 86 should be better justified ff.
> >- [...] or alludes to the convergence of curriculum, which was not proved
>
> We add a sketch of the convergence proof of the objective in line 80 as well as $J(\psi)$ to the 'global' comment of the rebuttal (please note that we used the old notation for the curriculum to make it accessible to other reviewers). We add a detailed proof to the appendix in the final version of the paper.
>
> We agree with the reviewers and 1) explain the equations 2) add references and 3) replace colloquial terms with technical statements. For 1) and 2) we refer to the global comment. Regarding 3):
> - The statement 'capacity [...] is exhausted' refers to the iteration where the optimization in line 80 reaches a fixed point in $\theta$
> - The term 'representational capacity' refers to the model complexity. The statement 'samples that lie within the representational capacity' refers to samples where the expert is able to achieve high log-likelihood values
>
> >- There is no description of the objective $J(\psi)$ and its lower bound, showing how the individual terms promote desirable behavior.
>
> - We focussed on the description of the per-component objective $J_z$. We believe that this explanation of the objective is the most intuitive, as $J_z$ is similar to the single expert objective (Section 3.1) with an additional term $\log q(z| o, a)$. We will add a description explaining why $\log q(z| o, a)$ promotes the claimed behavior.
>
> >[...] the method requires setting several important hyperparameters ff.
>
> We argue that the only hyperparameter that needs to be tuned is the curriculum pacing (or entropy scaling) $\eta$. We promote future research in an automatic tuning of $\eta$, similar to the approach proposed in [1].
>
>
>
> > [...] a natural question [...] would be to ask about performance on discrete domains ff.
>
> We will add discrete domains as promising avenue for future research.
>
> We thank the reviewers for noticing errors and notational shortcomings in the paper. We have addressed them and made the necessary corrections. Moreover, the valuable feedback has immensely contributed to the improvement and credibility of our research.
>
> [1] Haarnoja, Tuomas, et al. "Soft actor-critic algorithms and applications."

---

> > ### Comment · Reviewer_kVDY · 2023-08-14
> > **Thank you for the response**
> >
> > I have read all the reviews and the Authors' answers, and I appreciate the detailed discussion and new experimental results.
> >
> > I would like to know the Authors' response to the point on the description of the method in Sections 5.1-5.4, including insights about the method (e.g., what policies $p_{\theta_z}$ were learned, how they were acquired during training, where all modes were discovered, how a distribution that mixes policies $p_{\theta_z}$ looks like etc.), see my original review (but with a word 'skill' replaced by a 'policy')
> >
> > Concerning the results from Figure 6, I found the Authors' response in this respect slightly confusing: (a) the metrics seem to increase with $N_z$; (b) If the Authors' hypothesis about the overfit is true, the choice of $N_z$ matters and it has to be selected either via expert knowledge or hyperparameter optimization.

---

> > > ### Author Response · Authors · 2023-08-17
> > > **Re: Thank you for the response**
> > >
> > > # Rebuttal
> > >
> > > > I have read all the reviews and the Authors' answers, and I appreciate the detailed discussion and new experimental results.
> > >
> > > We extend our sincere gratitude to the reviewers for their questions, allowing us the opportunity to provide further clarification.
> > >
> > > > I would like to know the Authors' response to the point on the description of the method ff.
> > >
> > > We would like to address each of the concerns raised by the reviewer, one by one.
> > >
> > > > [...] what policies were learned [...]
> > >
> > > Visualizing the policies $p_{\theta_z}$ learned by our algorithm is challenging when dealing with high-dimensional problems. Nevertheless, the Obstacle Avoidance task lends itself to informative visual representations. Take for instance Figure 2 in the supplementary rebuttal PDF, which displays the curricula $p(o,a|z)$ with distinct colors assigned to individual values of $z$. Given that each policy $p_{\theta_z}$ is trained using samples selected from $p(o,a|z)$, this visualization serves as a direct means to observe the specific policies that have been learned.
> > > For instance, Figure 2b shows that the blue color corresponds to a policy guiding the robot's movement from the top left to the bottom right, the green color indicates a policy facilitating movement from the bottom left to the top right, and the orange color signifies a policy where the robot remains stationary.
> > >
> > >
> > > > [...] how they were acquired during training [...]
> > >
> > > We initialize the policy parameters randomly. Throughout the training process, the inclusion of the $\log q(z|o,a)$ term in our objective function ensures that curricula focus on distinct subsets of data. This specialization consequently results in the training of policies that align with these subsets. By employing a suitable number of expectation-maximization (EM) steps, the curricula converge, as demonstrated in the newly added proof, yielding the final policies. These converged curricula are visualized in Figure 2 of the supplementary rebuttal PDF.
> > >
> > > We incorporate an additional figure in the final version of the paper, which will illustrate the curricula at various stages, i.e., before training, during training, and after convergence. Similar to the format of Figure 2, this will provide a clearer visualization of how the curricula and policies evolve throughout the training process.
> > >
> > >
> > > > [...] where all modes were discovered [...]
> > >
> > > It is difficult to assess whether all modes in a data distribution are discovered as the number of modes is typically unknown. We, therefore, chose tasks such that the number of modes is known a-priori. This allows us to quantify the mode coverage using the 'entropy' metric. In response to the reviewer's suggestion, we will provide a more detailed explanation of how entropy is calculated in the main body of the paper, providing a clearer understanding of how we measure mode discovery. Please note that knowing the number of modes is not a requirement of our method but is merely used for evaluation.
> > >
> > > > [...] how a distribution that mixes policies looks like etc. [...]
> > >
> > > We are unsure of the reviewer's definition of 'mixing'. In the following, we assume that the reviewer refers to selecting different policies as mixing.
> > >
> > > The distribution that mixes policies in a MoE policy is the gating $p(z|o)$ which is approximated by $g_{\phi}(z|o)$. Given an oberservation $o'$, a sample $z' \sim g_{\phi}(z|o')$ is drawn to select a policy $p_{\theta_{z'}}(a|o',z')$.
> > >
> > > > [...] I found the Authors' response in this respect slightly confusing: (a) the metrics seem to increase with $N_z$
> > > ; (b) If the Authors' hypothesis about the overfit is true, the choice of $N_z$ matters, and it has to be selected either via expert knowledge or hyperparameter optimization.
> > >
> > > We apologize for any confusion that might have arisen from our earlier response concerning the issue of overfitting. We want to clarify that we do not consider overfitting due to a high number of components as a significant concern and it is also not visible in the evaluations of Figure 6. As can be seen, there are only small fluctuations in the performance of less than 2.5 percent for the Obstacle Avoidance task which are within the error bars and therefore not significant.
> > > In support of this assertion, we have conducted experiments using both the Obstacle Avoidance and Franka Kitchen tasks, employing 100 components over 5 separate seed runs. We could not observe any drop in performance and the observed success rates are within the error bars of the performance with 50 components.
> > > Hence, the only downside of choosing a large number of components is an increased computational burden and neither expert knowledge nor hyperparameter optimization is necessary to determine a suitable value for $N_z$.
> > >
> > > We appreciate the opportunity to clarify this matter and apologize for any confusion caused by our earlier response. We will include an ablation study where we use substantially more components in the camera-ready version of the paper.

---

> > > > ### Comment · Reviewer_kVDY · 2023-08-18
> > > >
> > > > Thank you for your explanations and for engaging in the discussion. Below are further comments and clarification questions.
> > > > * [What policies are learned] This is the explanation I would like to see in the paper. As a nitpick, it is unclear what "top left" or "bottom left" refers to when discussing obstacle avoidance since the robot seems to be starting from the same point on the left-hand side.
> > > > * [How a distribution that mixes policies looks] The distribution induced by the policy as described in lines 149-150 is a mixture distribution (mixes the distributions given by per component policy $p_{\theta_z}$, according to weights given by $g_\phi$). So the question concerns the values of weight given to every policy $p_{\theta_z}$? Is this distribution concentrated only on a few components, or maybe more uniform? Is there a reasonable explanation for that? Is this what one would expect? etc.
> > > > * [A follow-up question] $g_\phi$ minimizes the KL distance as defined in line 148. However, $p$ in this formula is conditioned on $(a, o)$, but $g_\phi$ is only conditioned on $o$. Is this a typo? If not, then it would be nice to see the Authors' explaining this choice.

---

> > > > > ### Author Response · Authors · 2023-08-18
> > > > >
> > > > > We express our gratitude to the reviewer for the insightful discussion. We are committed to addressing the remaining questions and comments.
> > > > >
> > > > > > [What policies are learned] This is the explanation I would like to see in the paper. As a nitpick, it is unclear what "top left" or "bottom left" refers to when discussing obstacle avoidance since the robot seems to be starting from the same point on the left-hand side.
> > > > >
> > > > > We're glad to hear that the provided explanation is helpful in enhancing the understanding of our approach. We will incorporate this into the camera-ready version of the paper.
> > > > >
> > > > > > [How a distribution that mixes policies looks] The distribution induced by the policy as described in lines 149-150 is a mixture distribution (mixes the distributions given by per component policy $p_{\theta_z}$, according to weights given by $g_{\phi}$). So the question concerns the values of weight given to every policy $p_{\theta_z}$? Is this distribution concentrated only on a few components, or maybe more uniform? Is there a reasonable explanation for that? Is this what one would expect? etc.
> > > > >
> > > > > We conducted experiments on the Obstacle Avoidance and Franka Kitchen task where we analyzed $p(z)$ and the entropy $\mathcal{H}(p(z))$ for a different number of components $N_z$. We observe that every component is given weight, i.e., $p(z) > 0 \forall z$, and that the entropy consistently demonstrates a monotonically increasing trend with increasing $N_z$. Additionally, we looked at the gating network for multiple observations $o'$, that is, $g_{\phi}(z|o')$ with $N_z=50$. We found that between $1-4$ components are given weight for every $o'$. This aligns with our expectation, since a) all components specialize on a subset of data as $p(z) > 0 \forall z$ and b) components specialize on different subsets since the number of components that are active per observation is much less than the total number of components $N_z$. We will put the quantitative results in the appendix of the camera-ready version.
> > > > >
> > > > > > [A follow-up question] $g_{\theta}$ minimizes the KL distance as defined in line 148. However, $p$ in this formula is conditioned on $(a,o)$, but $g_{\phi}$ is only conditioned on $o$. Is this a typo? If not, then it would be nice to see the Authors' explaining this choice.
> > > > >
> > > > > Indeed, it is not a typo. It is, however, mathematically equivalent to conditioning $p$ only on $o$ as shown below.
> > > > >
> > > > > $$
> > > > > \\min_{\phi} \mathbb{E}\_{o \sim p(o)}D\_{\text{KL}}(p(z|o)\|g_{\phi}(z|o))
> > > > > = \max_{\phi} \int_{\mathcal{O}}\sum_z p(o,z) \log {g_{\phi}(z|o)} \text{d}o
> > > > > = \max_{\phi} \int_{\mathcal{O}} \int_{\mathcal{A}} \sum_z p(o,a,z) \log {g_{\phi}(z|o)} \text{d}a \text{d}o
> > > > > = \max_{\phi} \int_{\mathcal{O}} \int_{\mathcal{A}} p(o,a) \sum_z p(z|o,a) \log {g_{\phi}(z|o)} \text{d}a \text{d}o = \\min_{\phi} \mathbb{E}\_{(o,a) \sim p(o,a)}D\_{\text{KL}}(p(z|o,a)\|g_{\phi}(z|o)).
> > > > > $$
> > > > >
> > > > > The reason for conditioning on $a$ is that we have access to $p(z|o,a)$ via Bayes' rule using the curricula, but not to $p(z|o)$. We will add this derivation to the appendix of the paper to prevent any confusion that might arise from our formulation.

---

### Author Rebuttal · Authors · 2023-08-09

We express our gratitude to the reviewers for their valuable suggestions and constructive feedback. Here, we post additions that might be of interest for all reviewers:

We have included a proof sketch, outlining the convergence of the algorithm proposed in our work, providing a more thorough understanding of its theoretical foundations.

Additionally, we have incorporated a comprehensive comparison between our proposed approach and the work introduced by Li et al. (2023) [1].

Lastly, we added a supplementary PDF file which contains
1. further quantitative and qualitative comparisons to the work of Li et al.,
2. a visual demonstration of the curricula of IMC for a different number of components on the obstacle avoidance task,
3. a figure that accompanies the convergence proof and visualizes the lower bound over training iterations.


We firmly believe that these additions contribute to elevating the overall quality and depth of our work. Once again, we thank the reviewers for their insights, which have played an instrumental role in refining our manuscript. We encourage the reviewers to reach out without hesitation if they have any additional questions or concerns.



## IMC Convergence Proof (Sketch)
**Singe Expert Objective**
$f(p(\mathcal{D}), \theta) = \mathbb{E}_{p(\mathcal{D})}[ \log p(a|o;\theta)] + \eta \mathcal{H}(\mathcal{D})$.
We perform coordinate ascent on $f$ which is guaranteed to converge to a stationary point if updating each coordinate results in a monotonic improvement of $f$ [2]. For fixed expert parameters $\theta$ we can find the unique $p(\mathcal{D})$ that maximizes $f$ [3] (see Section 3.1) and hence we have
$f(p(\mathcal{D})^{(i)}, \theta) \geq f(p(\mathcal{D})^{(i-1)}, \theta),$
where $i$ denotes the iteration.
Under suitable assumptions ($f$ is differentiable, its gradient is $L$-Lipschitz, and the learning rate $\alpha$ is chosen such that the descent lemma [4] holds), it holds that
$
f(p(\mathcal{D}), \theta^{(i)}) \geq f(p(\mathcal{D}), \theta^{(i-1)}),
$
when updating $f$ using gradient ascent. Hence, we are guaranteed to converge to a stationary point of $f$.

**Mixture of Experts Objective** $J(\psi)$.
To show that $J(\psi)$ converges to a stationary point, that is $\nabla_{\psi}J(\psi)=0$ we only have to show that $L(\psi^{(i)}, q) \geq L(\psi^{(i-1)}, q)$ as we tighten the lower bound in every E-step [5]. We again perform a coordinate ascent on  $L(\psi^{(i-1)}, q)$. In order to prove convergence we show that we have monotonic improvement on $L$ in every coordinate. It can easily be seen that $L$ is a maximum entropy objective with respect to $p(z)$ and $p(\mathcal{D}|z)$.
Hence, we can find the unique $p(z)$ and $p(\mathcal{D}|z)$ that maximize $L$ and thus have monotonic improvement. Noting that $q$ is not dependent on $\theta_z$, we can show monotonic improvement by using the same argument as for the single expert objective (the objective for $\theta$ is equal to the per-component objective for $\theta_z$).

**Additional Note.** While stochastic gradient ascent doesn't ensure strictly monotonic improvement in $L$ for $\theta_z$, our empirical observations (see accompanying PDF figure) reveal that $L$ indeed tends to increase monotonically in practice.

## Comparison: IMC vs. ML-Cur

We tested ML-Cur on the experiments used in the paper. We tested ${10, 30, 50}$ components. Furthermore, we did a hyperparameter sweep for the entropy scaling $\alpha$. We report the average performance $\pm$ one standard deviation across 10 seeds:

|  | Obstacle Avoidance |  |
| --- | --- | --- |
|  | success rate | entropy |
| IMC | $0.855 \pm 0.053$ | $0.930 \pm 0.031$ |
| ML-Cur | $0.454\pm 0.223$ | $0.035\pm 0.024$ |

|  | Block Pushing |  |  |
| --- | --- | --- | --- |
|  | success rate | entropy | distance error |
| IMC | $0.521 \pm 0.045$ | $0.654 \pm 0.041$ | $0.120 \pm 0.014$ |
| ML-Cur | $0.000 \pm 0.000$ | $0.000 \pm 0.000$ | $0.408 \pm 0.030$ |

|  | Table Tennis |  |
| --- | --- | --- |
|  | success rate | distance error |
| IMC | $0.870 \pm0.017$ | $0.153 \pm 0.007$ |
| ML-Cur | $0.836 \pm 0.020$ | $0.181 \pm 0.011$ |

For the Franka Kitchen task, we report the average performance for 1-4 tasks solved in the brackets.

|  | Franka Kitchen |  |
| --- | --- | --- |
|  | success rate | entropy |
| IMC | $[0.996, 0.969, 0.884, 0.626]$ | $[0.619, 1.037, 1847, 2.147]$ |
| ML-Cur | $[0.394, 0.000, 0.000, 0.000]$ | $[0.118, 0.000, 0.000, 0.000]$ |

In the accompanying PDF, we have incorporated supplementary quantitative and qualitative comparisons to further enhance the comprehensiveness of our analysis.

The results indicate the ML-Cur fails in the step-based setting (i.e., where the policy has to map from observations to actions, in contrast to the episodic tasks considered in Li, et al, where the policy maps from low-dimensional contexts to movement primitive parameters).
We hypothesize that linear experts and Gaussian curricula cause errors to accumulate across steps, leading to failure. This insight elucidates ML-Cur's struggles in manipulation tasks requiring precise actions.


[1] Li, et al. Curriculum-based imitation of versatile skills. arXiv preprint arXiv:2304.05171, 2023.

[2] Boyd, Stephen P., and Lieven Vandenberghe. Convex optimization. Cambridge university press, 2004.

[3] S. Levine. Reinforcement learning and control as probabilistic inference: Tutorial and review.
arXiv preprint arXiv:1805.00909, 2018.

[4] Avriel, Mordecai. "Nonlinear programming." Mathematical Programming for Operations Researchers and Computer Scientists. CRC Press, 2020. 271-367.

[5] Bishop, Christopher M., and Nasser M. Nasrabadi. Pattern recognition and machine learning. Vol. 4. No. 4. New York: springer, 2006.

---

### Author Response · Authors · 2023-08-21
**Brief Summary**

We again thank all reviewers for their time and effort in reviewing our paper and engaging in subsequent discussions. We are glad that three out of the four reviewers have voted in favor of acceptance. We would like to reiterate the main concerns and lines of discussion during the review.

**1. Structure and Notation.** We again thank the reviewers, in particular Reviewer kVDY, for their valuable insights to improve the structure and notation of the paper. We will include the points outlined in our response.

**2. Presentation of Results.** We thank Reviewer kVDY for their suggestions for improving the experiment section by providing more insights about the method. We will include the visualizations and ablation studies discussed during the rebuttal. These additions will illustrate the acquired individual policies and curricula, highlight the utilization of different components to address tasks, and provide insight into the mixing behavior of the gating distribution between policies.

**3. Hyperparameter Selection.** Reviewer kVDY raised the concern that our method requires knowing the number of modes in the data distribution a-priori for setting the number of components $N_z$, rendering the method hard to use in practice. We addressed this by clarifying that there is no one-to-one mapping between modes and $N_z$. Furthermore, we performed experiments to show that the only downside of choosing a large number of components is an increased computational burden. We, therefore, claim that neither expert knowledge nor hyperparameter optimization is necessary to determine a suitable value for $N_z$.

**4. Technical Details.** We address all technical concerns of Reviewer kVDY. We will improve the motivation of our objective by adding references, replacing informal terms with technical statements, and adding a convergence proof for our method as outlined in the rebuttal. Furthermore, we will add further technical details on why our objective promotes the claimed behavior.

**5. Benchmark tasks.** We thank Reviewer EzYZ for suggesting testing our method on the RoboMimic datasets. However, we believe that these tasks are not well suited to evaluate a method's ability to learn versatile skills as outlined in our response.

**6. Baselines.** We addressed Reviewer tAUb's concern by including a quantitative and qualitative comparison to the work of Li, et al (2023).

---

### Decision · Program_Chairs · 2023-09-21

**Decision:**

Accept (poster)

**Comment:**

The paper proposes a novel method for learning versatile robot skills that exhibit multiple modes. A curriculum based mixture of expert policy is used to focus on subsets of the training data. The approach is evaluated in simulated robotics tasks and compared against baselines.

The tackled problem is interesting and timely, and the method shows favorable results in the experiments. The reviewers initially also pointed out some shortcomings including: writing issues, missing and unclear details, too limited experimental evaluations (baselines and tasks), missing theoretical backing, and connection to related work.

The authors provided additional results and convincing arguments in the rebuttal phase. Those managed to address all (major) concerns of the reviewers, and they are now all recommending accepting this paper.